# Text-Driven Fusion for Infrared and Visible Images: Achieving Image Scene Adaptation on Hyperbolic Space

Huan Kang[1]  Hui Li[1]  Tianyang Xu[1]  Tao Zhou[1]  Xiao-Jun Wu[1]  Josef Kittler[2]

## Abstract

Infrared and visible image fusion aims to integrate complementary modalities, while existing Euclidean methods impose rigid distance metrics that distort multi-modal interactions and parent-to-child semantic hierarchies. To overcome these limitations, we introduce a text-driven fusion framework empowered by hyperbolic manifold learning. During training, BLIP-extracted text prompts serve as topological anchors within the hyperbolic space, guiding vision-attribute alignment through hyperbolic embeddings that naturally accommodate varying semantic granularities. By exploiting the exponential volume growth dictated by the Poincaré ball's negative curvature, this approach seamlessly embeds hierarchical trees to encode coarse-to-fine semantics without metric saturation, while the vast peripheral space prevents texture distortion during cross-modal fusion. At inference, the fusion process autonomously adapts to input content using the learned text-attribute priors, completely eliminating the need for textual input. Experimental results show our method outperforms state-of-the-art approaches on benchmark datasets, with code available at https://github.com/Shaoyun2023/TEDFusion.

## 1. Introduction

Image fusion plays a crucial role in the computer vision community, addressing the growing need for comprehensive multi-source image processing in practical applications (Ma et al., 2019a; Xu et al., 2020; Li et al., 2017). A single image

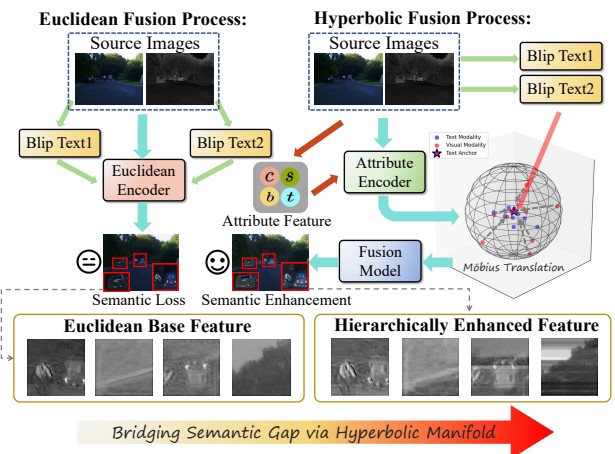

*Figure 1.* Breaking free from the rigid geometry of Euclidean space, our text-driven hyperbolic framework bridges the modality gap. It leverages semantic anchors to hierarchically align and enhance cross-modal features, achieving a richer representation of the scene semantics.

source is often insufficient for analyzing and recognizing complex scenes. Advances in sensor technology have enabled the acquisition of multi-band, multi-perspective, and multi-resolution image data, providing a rich foundation for fusion tasks. Specifically, the primary goal of image fusion is to integrate information from multiple sources to produce a more informative, complete, and reliable composite image. Among the fusion tasks, Infrared and Visible Image Fusion (IVIF) stands out as a prominent example. It combines the thermal radiation information from infrared images with the textual details of visible images, resulting in a fused image that retains both thermal characteristics and clear textures. This technology has been widely applied in diverse fields such as military reconnaissance (Liu et al., 2022), target tracking (Li et al., 2020a; Long Li et al., 2019; Li et al.) and medical diagnosis (Tang et al., 2022b; Xu & Ma, 2021).

Visual information is encoded in the spatial domain of an image, where object positions, shapes, and layout relationships provide essential cues for fusion. Aligning such structural information across modalities, such as infrared and visible, is critical, yet challenging due to their complementary characteristics and distribution differences. Text, as a discrete

[1]School of Artificial Intelligence and Computer Science, Jiangnan University, Wuxi, China [2]Centre for Vision, Speech and Signal Processing (CVSSP), University of Surrey, Guildford, UK. Correspondence to: Xiao-Jun Wu <wu_xiaojun@jiangnan.edu.cn>.

*Proceedings of the 43rd International Conference on Machine Learning*, Seoul, South Korea. PMLR 306, 2026. Copyright 2026 by the author(s).

symbolic modality, conveys high-level semantics through linguistic constructs. It can offer contextual guidance to aid fusion (Yi et al., 2024; Cheng et al., 2025b; Zhang et al., 2024; Wang et al., 2024a). However, many existing fusion methods depend heavily on textual input being provided during both training and inference, which introduces significant complexity and operational constraints. This reliance, coupled with the inherent gap between visual signals and semantic symbols, prevents accurate cross-modal matching in standard Euclidean networks, thereby resulting in weak semantic interaction and inefficient feature integration.

A fundamental limitation of image fusion methods, which are invariably Euclidean-based, stems from their reliance on flat embedding spaces for cross-modal alignment, where all embedded features are processed uniformly under the same distance metric (Murphy, 2012). This uniform treatment becomes particularly problematic when modelling multi-modal and multi-scale data. By learning image-text correlations in a Euclidean space, these methods often capture only superficial and rigid associations, failing to represent the inherent hierarchical semantics (Vendrov et al., 2015) or preserve the topological organization of visual and textual features (Huang & Van Gool, 2017; Kang et al., 2025a; Chen et al., 2024; Brooks et al., 2019; Kang et al., 2025b). For instance, high-level contextual features (*e.g.*, scene categories) become geometrically close to a broad set of unrelated features, while low-level details (*e.g.*, texture or edges) are only close to their immediate neighbors. During fusion, such flat geometry distorts semantic alignment and misrepresents the non-linear statistical correlations between modalities (Zhao et al., 2023; Li et al., 2025; Liu et al., 2023; Li & Wu, 2024; Liu et al., 2025). Consequently, the fusion process fails to achieve semantic consistency, often oversimplifying fine-grained details or introducing semantic contradictions (Cheng et al., 2025a; Zhu et al., 2024), which ultimately limit generalization in unseen scenarios and hinders deployment in real-time applications.

To tackle the challenges of scene information loss and inherent limitations of Euclidean feature modelling, we propose TEDFusion, a text-driven image fusion framework with deep semantic-scene alignment. Unlike methods that rely solely on low-level features (Li et al., 2023; Tang et al., 2023), TEDFusion leverages semantic guidance from text during training to enhance attribute-aware feature interaction across modalities, while during inference, it operates in a text-free manner, significantly simplifying user interaction. The core of our approach lies in bridging the semantic gap between modalities through hyperbolic modelling. As shown in Fig. 1, conventional Euclidean embedding spaces often impose linear constraints on feature relationships, which struggle to capture the inherently hierarchical nature of semantic-visual associations, resulting in suboptimal fusion with either detail loss or semantic inconsistency.

To overcome this, we project attribute features and text embeddings into a shared hyperbolic space (Ungar, 2008; Ganea et al., 2018a;b), specifically leveraging the negative curvature of the Poincaré ball to model hierarchical relationships naturally. In this geometrically structured space, text-derived semantics serve as anchors near the origin, organizing visual features progressively by abstraction level, thereby enabling semantically coherent fusion without being constrained by Euclidean compatibility. Furthermore, we propose a loss function on the hyperbolic manifold to facilitate learning this semantic relationship.

Our main contributions are summarized as follows:

- We have designed an adaptive image fusion framework that enhances critical scene attributes (*e.g.*, brightness, contrast, texture) under semantic guidance, effectively reducing information loss and misalignment in fusion.

- Being supervised by text semantics during training, our model derives its own image-type-dependent fusion criteria. During inference, it operates entirely text-free, dynamically adjusting the fusion strategy based solely on the visual input for superior scene adaptation.

- By embedding features in a hyperbolic manifold, we bridge the modality gap existing in the Euclidean space. Our hyperbolic loss enforces a more native, as well as hierarchical alignment between semantics and text, leading to better consistency with human perception.

## 2. Related Works

### 2.1. Infrared-Visible Image Fusion Methods

Traditional IVIF methods primarily include approaches based on sparse representation (Li et al., 2023), subspace (Zhang et al., 2021; Li et al., 2020b), and multi-scale transformations (Li et al., 2018). Although these methods have achieved good fusion results within certain limits, they still have some drawbacks. Most are built on specific mathematical models, which may limit their adaptability and flexibility across diverse scenarios and fusion tasks, making it difficult to fully meet the requirements for efficient adaptation to various fusion needs.

Deep learning has revolutionized IVIF through automatic feature learning, primarily via CNNs (Tang et al., 2023; Xu et al., 2020; Li & Wu, 2018; Wang et al., 2024b; Liu et al., 2024), GANs (Ma et al., 2020; 2019b) and Transformers (Zhao et al., 2023; Li & Wu, 2024; Qu et al., 2022; Cheng et al., 2025a). CNNs extract and integrate features from local patterns, GANs generate natural fused images through adversarial training, and Transformers (Vaswani, 2017) capture global dependencies via self-attention. However, these methods lack high-level semantic understanding, limiting

performance in complex scenarios.

In recent years, visual-language models (Lin et al., 2024; Li et al., 2022; Zhou et al., 2022) have advanced rapidly, demonstrating powerful cross-modal understanding capabilities by effectively learning relationships between visual and linguistic information. These models are pre-trained on large-scale datasets and exhibit strong performance across various vision-language tasks, such as Visual Question Answering (VQA) (Zhou et al., 2020; Gupta & Kembhavi, 2023; Zeng et al., 2024), image captioning, and visual reasoning.

The introduction of textual guidance in image fusion stems from the need for semantic-aware integration (Cheng et al., 2025b; Yi et al., 2024; Wang et al., 2025; 2024a). Unlike conventional methods that process all scenes uniformly, text-driven approaches condition the fusion process on high-level scene descriptions. This allows adaptive feature prioritization, where fusion strategies can be dynamically tailored to the semantic context. However, the clash between continuous image features and discrete linguistic semantics often prevents deep alignment. We posit that the inherent hierarchical and tree-like structure underlying both visual scenes and linguistic semantics is key to bridging this gap.

## 2.2. Hyperbolic Manifold

Hyperbolic neural networks (Ganea et al., 2018b) leverage the structural properties of hyperbolic spaces, such as exponential volume expansion, to effectively embed hierarchical data. This framework facilitates the development of hyperbolic deep learning layers, demonstrating superior performance over Euclidean methods in modelling tree-like structures. Building on this, the hyperbolic approach has been successfully applied to natural language processing (Dhingra et al., 2018; He et al., 2024), where its capacity to capture linguistic hierarchies across different levels of abstraction, including word semantics and phrase composition, has been validated through unsupervised learning on text corpora. The methodology further extends to computer vision (Khrulkov et al., 2020; Kwon et al., 2024), proving particularly effective in modelling the complex hierarchies inherent in visual scenes. This geometrically grounded representation allows for a more natural organization of visual concepts, from broad categories down to fine-grained details. By adapting standard network architectures to accommodate this non-Euclidean geometry, hyperbolic methods significantly enhance tasks such as image classification through their innate ability to represent nested structural relationships.

More recently, hyperbolic manifolds have been adopted for vision-text representation learning (Pal et al., 2025; Poppi et al., 2025; Desai et al., 2023). A central focus is to capture the hierarchical entailment between images and text in a shared embedding space, which provides a geometrically natural way to model semantic granularity and compositionality across modalities. Within this joint space, cross-modal concepts can be effectively organized and balanced based on their specific levels of semantic depth.

These recent advancements underscore the considerable potential of hyperbolic geometry in IVIF. Its negative curvature provides a naturally expanding and continuous semantic space, which enables more effective alignment and interaction between cross-modal features without suffering from representation saturation. This geometric property facilitates smoother integration of heterogeneous data, thereby helping to bridge the cross-modal semantic gap and ultimately achieve more coherent and detail-preserving fusion.

## 3. The Proposed Method

### 3.1. Text-Driven Fusion Framework

Our network (Fig. 2) integrates a deep visual encoder and a BLIP-based textual encoder (Li et al., 2022). A registered image pair $I_{ir}$ and $I_{vi}$ is processed by the deep encoder $\mathcal{F}_e(\cdot)$, which uses ViT's attention mechanisms (Dosovitskiy et al., 2021) to extract and concatenate common and unique features. For adaptive scene enhancement, we propose the Attribute and Text Gating Fusion Module (ATGFM). It injects attribute features into the text stream to bolster cross-modal learning, followed by semantic-guided weighting of multi-scale features. Subsequently, the Text-Image Association Module (TIAM) learns hierarchical features linking images and text, constrained by a hyperbolic loss. To reconstruct the final fused image $I_f$, the decoder $\mathcal{F}_d(\cdot)$ adopts the multi-scale Transformer block components following Text-IF (Yi et al., 2024). Crucially, to effectively aggregate TIAM's hierarchical semantics, our nested architecture leverages dense cross-scale interactions, seamlessly fusing high-level cross-modal alignments with low-level visual details.

### 3.2. Attribute and Text Gating Fusion Module

The textual domain $\mathcal{T}$ provides a structured semantic framework to complement image attributes $\mathcal{A}$, where ambiguous visual features are disambiguated via textual guidance, while image attributes adaptively refine textual representations through bidirectional cross-modal enhancement.

We extract four attribute pairs from infrared and visible modalities for multi-modal alignment. For visual perception, saturation $S_u$ (from HSV color space) and brightness $B_r$ (V-channel) quantify color distribution and illumination intensity. Structural attributes are encoded via multi-scale Gray-Level Co-occurrence Matrix (GLCM) (Haralick & Dinstein, 2007; Singh et al., 2017): energy $E$ and homogeneity $H$ are linearly combined with modality-specific

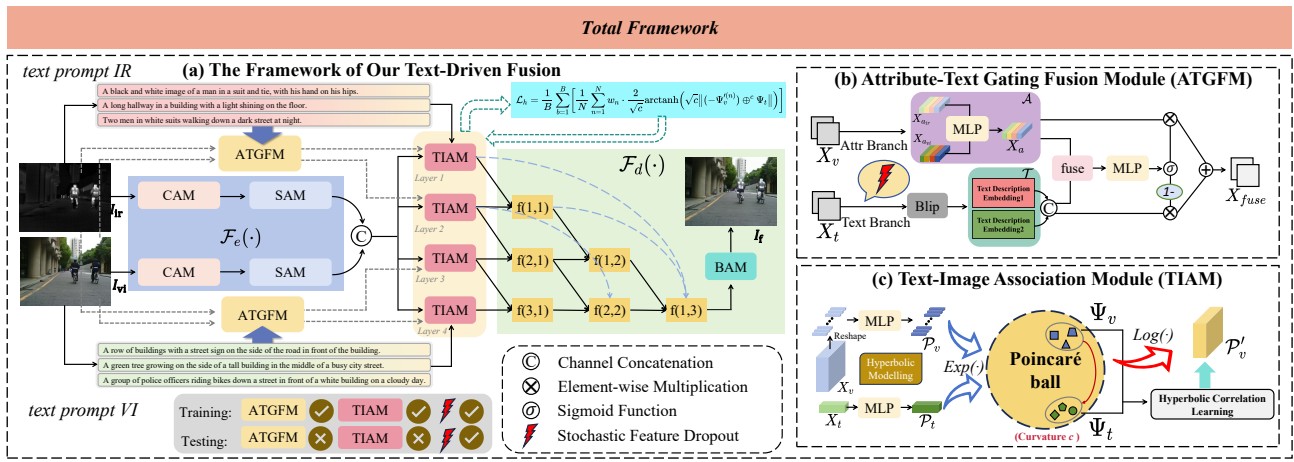

*Figure 2.* The workflow of our TEDFusion. (a) illustrates the main architecture of the network, including a Transformer-based feature extraction layer and a nested decoding network. (b) details the workflow of the attribute-text gating fusion module, which integrates text-related scene semantics and embeds them into the network. (c) illustrates the schematic of text-image feature association in hyperbolic space, achieving a geometric semantic reassignment through this process.

weights to model texture complexity $T_c = w_1 E + w_2 H$, while contrast $C_s$ (computed as standard deviation across spatial channels) constrains intensity variations. More details can be found in Appendix A.

In the Attribute and Text Gating Fusion Module (ATGFM), the attribute set $[S_u, B_r, T_c, C_s]$ extracted from original visual features $X_v$ is divided into infrared and visible parts $X_{a_{ir}}$ and $X_{a_{vi}}$, which are concatenated and projected through a shared MLP to produce a unified attribute embedding $X_a$. Meanwhile, the text features $X_t$ are encoded by BLIP into modality-specific descriptions, which are then concatenated and projected into a 64-dimensional space. For each attribute-text pair, a gating network $g(\cdot) = \sigma(\text{MLP}([X_a, X_t]))$ dynamically balances their contributions through $X_{fuse} = g \odot X_a + (1 - g) \odot X_t$. During training, stochastic dropout (30% probability) on text embeddings forces the model to learn robust attribute-text correspondences. During inference, when textual descriptions are unavailable, the learned gating weights guide attribute embeddings to functionally substitute text features, preserving semantic consistency while avoiding fusion artifacts across infrared-visible domains.

### 3.3. Text and Image Association Module

The hyperbolic semantic alignment module operates on two input modalities: the image feature tensor $X_v \in \mathbb{R}^{B \times C \times H \times W}$ and the text feature vector $X_t \in \mathbb{R}^{B \times D}$, which is obtained by projecting raw text features through an MLP. Here, $B$ denotes the batch size, $C$ and $D$ represent the channel dimensions of image and text features respectively, and $H \times W$ indicates the spatial resolution of the feature maps.

Let the image and text features after linear projection be $\mathcal{P}_v = \text{MLP}(\text{Reshape}(X_v)) \in \mathbb{R}^{B \times N \times d_h}$ and $\mathcal{P}_t = \text{MLP}(X_t) \in \mathbb{R}^{B \times d_h}$, where $N = H \times W$ represents the number of spatial patches, and $d_h$ is the target dimension of the hyperbolic embedding space. These projected vectors reside in the tangent space at the origin $\mathbf{0}$ of the Poincaré ball, denoted as $\mathcal{T}_0 \mathbb{D}_c^{d_h}$, where $\mathbb{D}_c^{d_h} = \{\mathbf{x} \in \mathbb{R}^{d_h} : \sqrt{c}\|\mathbf{x}\| < 1\}$ represents the Poincaré ball of dimension $d_h$, and $c$ is the curvature parameter of the hyperbolic space.

The exponential mapping operation then projects these tangent vectors onto the Poincaré ball manifold to obtain $\Psi_v$ and $\Psi_t$:

$$
\begin{aligned}
\Psi_t &= \exp_\mathbf{0}^c(\mathcal{P}_t) = \tanh\left(\sqrt{c}\|\mathcal{P}_t\|\right) \cdot \frac{\mathcal{P}_t}{\sqrt{c}\|\mathcal{P}_t\| + \epsilon}, \\
\Psi_v &= \exp_\mathbf{0}^c(\mathcal{P}_v) = \tanh\left(\sqrt{c}\|\mathcal{P}_v\|\right) \cdot \frac{\mathcal{P}_v}{\sqrt{c}\|\mathcal{P}_v\| + \epsilon},
\end{aligned}
\tag{1}
$$

where $\epsilon$ is a small constant for numerical stability, and $\|\cdot\|$ denotes the Euclidean norm.

Subsequently, a Möbius translation (Vermeer, 2005) relocates the image features relative to the textual anchor point:

$$
\Psi_v' = \frac{(1 + 2c\langle\Psi_v, -\Psi_t\rangle + c\| - \Psi_t\|^2)\Psi_v + (1 - c\|\Psi_v\|^2)(-\Psi_t)}{1 + 2c\langle\Psi_v, -\Psi_t\rangle + c^2\|\Psi_v\|^2\| - \Psi_t\|^2}.
\tag{2}
$$

To leverage multi-scale visual information, we extract features at $N = 4$ scales and apply the above hyperbolic transformations independently at each scale, yielding $\{\Psi_v'^{(n)}\}_{n=1}^N$ while keeping $\Psi_t$ shared across scales. The adaptive hyperbolic loss function then measures the quality

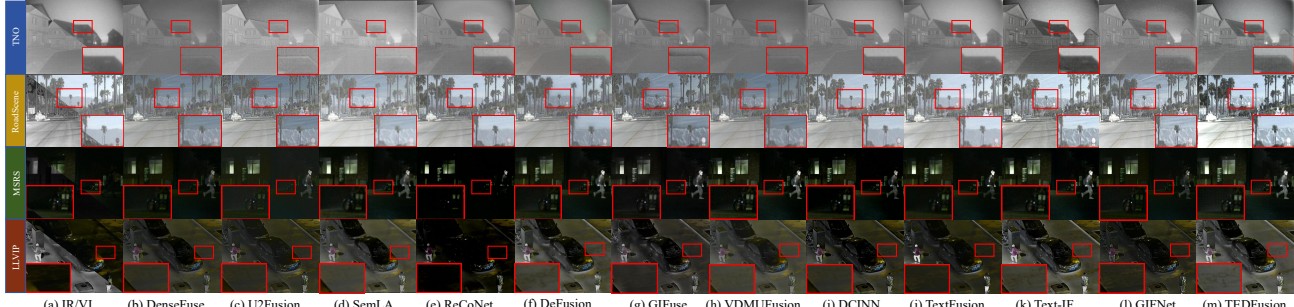

(a) IR/VI   (b) DenseFuse   (c) U2Fusion   (d) SemLA   (e) ReCoNet   (f) DeFusion   (g) GIFuse   (h) VDMUFusion   (i) DCINN   (j) TextFusion   (k) Text-IF   (l) GIFNet   (m) TEDFusion

*Figure 3.* A quantitative comparison of performance of the IVIF task. The displayed examples are from the TNO, RoadScene, MSRS, and LLVIP datasets.

*Table 1.* Quantitative experiments on the TNO, RoadScene, MSRS and LLVIP datasets. **Bold and underlined** indicates the best, whereas red indicates the second best.

| | TNO Dataset | | | | | RoadScene Dataset | | | | | MSRS Dataset | | | | | LLVIP Dataset | | | | |
|---|---|---|---|---|---|---|---|---|---|---|---|---|---|---|---|---|---|---|---|---|
| | EN | SD | SF | AG | SCD | EN | SD | SF | AG | SCD | EN | SD | SF | AG | SCD | EN | SD | SF | AG | SCD |
| DenseFuse (Li & Wu, 2018) | 6.31 | 23.47 | 3.84 | 1.65 | 1.67 | 6.77 | 31.38 | 5.21 | 2.23 | 1.42 | 5.93 | 23.55 | 6.02 | 2.05 | 1.25 | 6.67 | 29.95 | 8.11 | 2.46 | 1.24 |
| U2Fusion (Xu et al., 2020) | 6.77 | 32.39 | 6.70 | 3.02 | 1.78 | 6.84 | 32.66 | 7.54 | 3.28 | 1.38 | 5.21 | 22.67 | 8.06 | 2.52 | 1.15 | 6.58 | 33.47 | 11.30 | 3.46 | 1.33 |
| ReCoNet (Huang et al., 2022) | 6.87 | 37.70 | 5.26 | 2.27 | 1.77 | 6.98 | 40.72 | 6.44 | 2.69 | 1.58 | 4.23 | 41.71 | 9.98 | 3.00 | 1.26 | 5.26 | 39.13 | 8.88 | 2.69 | 1.39 |
| DeFusion (Liang et al., 2022) | 6.58 | 28.98 | 4.24 | 1.81 | 1.61 | 6.91 | 34.79 | 5.44 | 2.32 | 1.37 | 6.38 | 35.43 | 8.15 | 2.65 | 1.27 | 7.09 | 39.23 | 9.86 | 2.98 | 1.26 |
| SemLA (Xie et al., 2023) | 6.86 | 37.05 | 7.14 | 2.12 | 1.64 | 6.95 | 38.57 | 8.12 | 2.52 | 1.46 | 6.42 | 33.12 | 6.35 | 2.26 | 1.51 | 7.12 | 42.14 | 7.39 | 2.60 | 1.56 |
| GIFuse (Wang et al., 2024c) | 6.39 | 24.78 | 5.97 | 2.38 | 1.62 | 6.95 | 34.75 | 8.29 | 3.40 | 1.37 | 6.31 | 32.49 | 10.43 | 3.31 | 1.38 | 6.84 | 33.55 | 14.12 | 4.00 | 1.19 |
| VDMUFusion (Shi et al., 2024) | 6.32 | 23.63 | 4.28 | 1.75 | 1.67 | 6.80 | 32.24 | 5.67 | 2.36 | 1.42 | 5.95 | 23.60 | 6.49 | 2.24 | 1.27 | 6.74 | 31.03 | 8.71 | 2.74 | 1.28 |
| DCINN (Wang et al., 2024b) | 6.69 | 31.75 | 5.94 | 2.48 | 1.70 | 6.94 | 36.68 | 7.53 | 3.05 | 1.45 | 6.04 | 40.29 | 10.78 | 3.47 | 1.47 | 6.79 | 37.31 | 12.58 | 3.69 | 1.44 |
| TextFusion (Cheng et al., 2025b) | 6.97 | 41.55 | 6.41 | 2.62 | 1.68 | 6.98 | 40.49 | 6.70 | 2.64 | 1.53 | 6.23 | 40.85 | 10.32 | 3.04 | 1.56 | 6.65 | 37.30 | 11.22 | 3.12 | 1.22 |
| Text-IF (Yi et al., 2024) | **7.14** | 42.73 | 7.88 | 3.33 | 1.69 | **7.31** | 46.70 | 10.39 | 4.42 | 1.48 | 6.72 | **44.03** | 11.63 | 3.84 | **1.65** | 7.24 | 45.38 | 16.29 | 4.81 | 1.46 |
| GIFNet (Cheng et al., 2025a) | 6.93 | 38.57 | **8.36** | 3.22 | **1.87** | 7.24 | 46.34 | 10.78 | 4.29 | **1.77** | 5.94 | 32.90 | **12.71** | 3.37 | 1.41 | 6.72 | 37.33 | 17.39 | 4.63 | 1.45 |
| TEDFusion (Ours) | 7.09 | **44.52** | **10.90** | **3.88** | 1.83 | 7.07 | **60.85** | **13.88** | **4.79** | 1.75 | **6.74** | 42.90 | 12.22 | **4.12** | 1.66 | **7.36** | **48.75** | 15.70 | **4.83** | **1.69** |

of semantic alignment:

$$\mathcal{L}_h = \frac{1}{B}\sum_{b=1}^{B}\left[\frac{1}{N}\sum_{n=1}^{N} w_n \cdot \frac{2}{\sqrt{c}}\mathrm{arctanh}\left(\sqrt{c}\|(-\Psi_v'^{(n)}) \oplus^c \Psi_t\|\right)\right],$$ (3)

where $b$ is the batch index. The scale-specific adaptive weighting term is defined as:

$$w_n = \exp\left(-\frac{|\|\Psi_v'^{(n)}\|_{hyp} - \|\Psi_t\|_{hyp}|}{\tau}\right),$$ (4)

where $\tau$ is a temperature parameter controlling the sharpness of weight distribution and the hyperbolic norm $\|\mathbf{x}\|_{hyp}$ is the distance from the origin to $\mathbf{x}$ in the Poincaré ball, computed as $d_c(\mathbf{0},\mathbf{x}) = \frac{2}{\sqrt{c}}\mathrm{arctanh}(\sqrt{c}\|\mathbf{x}\|)$ for $\mathbf{x} \in \mathbb{D}_c^{d_h}$.

The exponentially expanding space naturally accommodates hierarchical visual semantics, while the curvature-adaptive formulation mitigates geometric conflicts between heterogeneous modalities. The adaptive hyperbolic loss further guides the optimization by prioritizing alignment in semantically relevant regions, ensuring the fused output preserves critical information from both spectra while maintaining cross-modal semantic consistency.

Finally, the hierarchically enhanced image feature $\Psi_v'$ in the hyperbolic space is projected back into the Euclidean space

via a logarithmic mapping to obtain the final hyperbolic-enhanced representation $\mathcal{P}_v'$, which facilitates the subsequent fusion network:

$$\mathcal{P}_v' = \log_{\mathbf{0}}^c(\Psi_v') = \frac{1}{\sqrt{c}} \cdot \frac{\mathrm{arctanh}(\sqrt{c}\|\Psi_v'\|)}{\|\Psi_v'\| + \epsilon}\Psi_v',$$ (5)

where $\mathcal{P}_v' \in \mathbb{R}^{B \times d_h}$ lies in the tangent space $\mathcal{T}_{\mathbf{0}}\mathbb{D}_c^{d_h}$ and can be processed by standard Euclidean layers.

### 3.4. Training Process and Loss Function

Given the challenges in IVIF (Ma et al., 2019b; 2021), we propose a hierarchical constraint strategy to progressively strengthen scene representation by exploiting cross-modal feature correlations.

First, a pixel-level constraint $L_{int}$ preserves fundamental intensity patterns by minimizing differences between the fused image and the element-wise maximum of source images:

$$L_{int} = \frac{1}{HW}\|I_f - \max(I_{ir}, I_{vi})\|_1.$$ (6)

A gradient-based constraint $L_{grad}$ captures texture details by comparing gradients:

$$L_{grad} = \frac{1}{HW}\||\nabla I_f| - \max(|\nabla I_{ir}|, |\nabla I_{vi}|)\|_1,$$ (7)

*Table 2.* An experimental comparison of different ablation settings. **Bold and underlined** denotes the best result, and red the second best.

|     | Variants | EN | SD | SF | AG | SCD |
|-----|----------|-----|-----|------|------|------|
| I | w/o Text | 7.05 | 43.68 | 7.95 | 3.23 | **1.85** |
| II | Text-only Training | 7.07 | 43.91 | 9.23 | 3.46 | 1.83 |
| III | w/o ATGFM | 7.06 | 44.17 | 9.30 | 3.39 | 1.81 |
| IV | TIAM-Euclidean | 6.99 | 44.24 | 10.05 | 3.21 | 1.80 |
| V | w/o $L_h$ | 6.97 | 42.70 | **12.41** | 3.31 | 1.81 |
| VI | c=0.5 | 6.97 | 43.19 | 9.40 | 3.85 | 1.81 |
| VII | c=0.7 | 7.00 | 43.47 | 9.58 | 3.74 | 1.82 |
| VIII | Ours (c=1.0) | **7.09** | **44.52** | 10.90 | **3.88** | 1.83 |

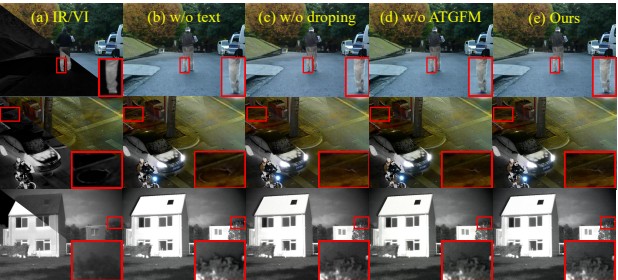

*Figure 4.* Visualization results of different ablation settings, where our proposed settings demonstrate the best performance.

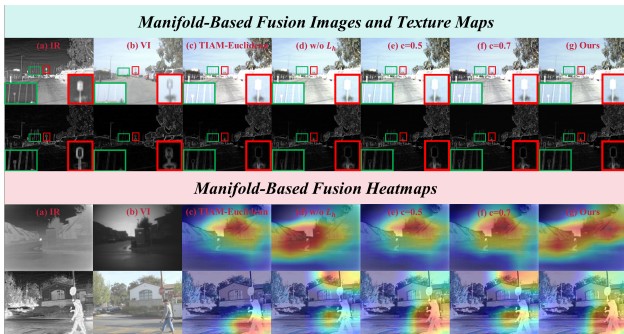

*Figure 5.* Visualization results of hyperbolic correlation learning. Our framework preserves fine-grained information while steering the network's attention toward critical regions.

where $\nabla$ denotes the Sobel gradient operator.

The SSIM loss (Wang et al., 2004) ensures structural integrity:

$$L_{ssim} = (1 - \text{SSIM}(I_f, I_{vi})) + (1 - \text{SSIM}(I_f, I_{ir})). \quad (8)$$

Additionally, we adopt the hyperbolic manifold-based loss $L_h$ introduced in Section 3.3 to model the hierarchical relationships in cross-modal features, leveraging the exponential expansion property of hyperbolic space for multi-scale semantic learning.

The total loss function is as follows:

$$L_{total} = L_{int} + \alpha L_{grad} + \beta L_{ssim} + \gamma L_h, \quad (9)$$

where $\alpha$, $\beta$, and $\gamma$ control hierarchical learning weights.

## 4. Experiments

### 4.1. Experimental Settings

**Datasets and Parameter Setting.** To comprehensively evaluate the performance of our method, we conduct experiments on four widely used and publicly available image fusion datasets: TNO (Toet & Hogervorst, 2012), RoadScene (Xu et al., 2020), MSRS (Tang et al., 2022a), and LLVIP (Jia et al., 2021). These datasets cover diverse scenarios, including nighttime surveillance, road scenes, and urban lighting, thereby providing a robust basis for assessing generalization capability. For downstream tasks such as object detection and semantic segmentation, we employ the MSRS and M3FD (Liu et al., 2022) datasets, which contain corresponding annotations for quantitative evaluation.

It is worth noting that our pretraining is performed on the MSRS dataset, and the model is trained end-to-end on an NVIDIA GeForce RTX 4070 Super GPU. The batch size is set to 2, and the learning rate is set to 0.0001 with the Adam optimizer. In addition, the entire framework is implemented in PyTorch.

**Compared Methods and Evaluation Metrics.** We compare the proposed method against eleven state-of-the-art image fusion approaches, including both well-established and recently published networks, to ensure a comprehensive evaluation. The competing methods are: DenseFuse (Li & Wu, 2018), U2Fusion (Xu et al., 2020), ReCoNet (Huang et al., 2022), DeFusion (Liang et al., 2022), SemLA (Xie et al., 2023), GIFuse (Wang et al., 2024c), VDMUFusion (Shi et al., 2024), DCINN (Wang et al., 2024b), TextFusion (Cheng et al., 2025b), Text-IF (Yi et al., 2024) and GIFNet (Cheng et al., 2025a).

For quantitative evaluation, we employ five full-reference and no-reference image quality metrics: Entropy (EN), which reflects the amount of information contained in the fused image; Standard Deviation (SD), measuring contrast and dispersion; Spatial Frequency (SF), characterizing overall image activity; Average Gradient (AG), reflecting edge preservation and clarity; and Sum of Correlation Difference (SCD), evaluating the transfer of complementary information from source images. A detailed explanation of these metrics can be found in (Ma et al., 2019a).

**Text Setting.** In this work, we use BLIP to automatically generate one caption per image on-the-fly during training and evaluation, without manual editing. For each batch, visible and infrared images are processed separately through BLIP to produce captions, yielding two text embeddings per sample that describe the visual content of each input.

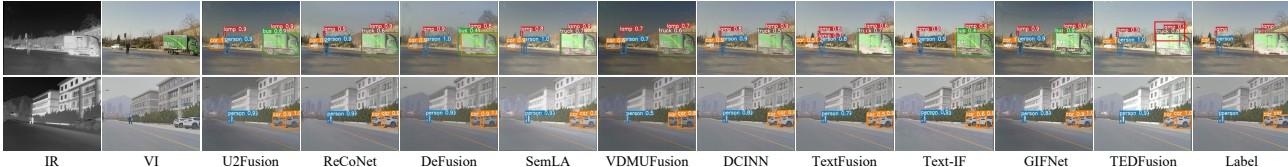

*Figure 6.* Visualization results of the object detection task on the M3FD dataset.

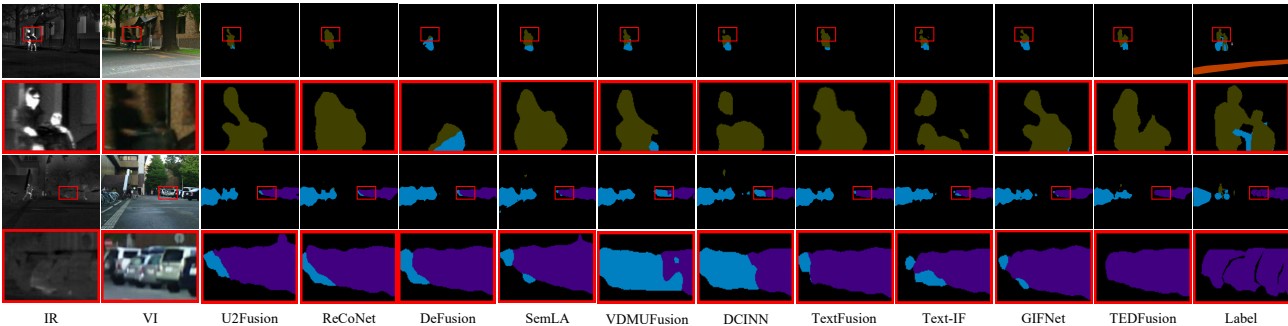

*Figure 7.* Visualization results of the semantic segmentation task on the MSRS dataset.

*Table 3.* Downstream experiments on the M3FD and MSRS datasets. **Bold and underlined** denotes the best result, and red the second best, respectively.

| | Object Detection | | | | | | | Semantic Segmentation | | | | |
|---|---|---|---|---|---|---|---|---|---|---|---|---|
| | Per | Car | Bus | Lamp | Motor | Truck | mAP@0.5 | Unl | Per | Car | Bik | mIoU |
| DenseFuse | **75.04** | 81.19 | 33.04 | 36.87 | 40.86 | 24.10 | 48.52 | 97.82 | 24.70 | 73.91 | **65.37** | 65.45 |
| U2Fusion | 71.92 | 81.49 | 31.27 | 43.83 | 35.89 | 42.91 | 51.22 | 97.70 | 34.44 | 70.47 | 60.56 | 65.79 |
| ReCoNet | 72.57 | 81.03 | 39.26 | 43.99 | 28.55 | **48.42** | 52.30 | 97.52 | 25.94 | 70.89 | 54.60 | 62.24 |
| DeFusion | 73.75 | **81.66** | 37.81 | 41.29 | **48.80** | 33.43 | **53.45** | 97.85 | 21.84 | 74.18 | **64.70** | 64.64 |
| SemLA | 66.86 | 75.89 | 33.64 | 39.88 | 19.15 | 22.48 | 42.98 | 97.84 | 32.96 | 73.87 | 59.95 | 66.16 |
| GIFuse | 53.19 | 74.62 | 18.34 | 21.91 | 2.19 | 12.52 | 30.59 | **98.02** | **40.73** | 75.54 | 61.29 | **68.90** |
| VDMUFusion | 57.58 | 76.07 | 13.80 | 46.56 | 34.55 | 27.28 | 42.64 | 97.80 | 32.92 | 73.10 | 59.07 | 65.73 |
| DCINN | 74.82 | 79.89 | 31.64 | 42.79 | 41.42 | 36.39 | 51.16 | 97.78 | 35.98 | 72.42 | 57.60 | 65.94 |
| TextFusion | 68.79 | 78.26 | 20.31 | 30.20 | 16.93 | 14.99 | 38.25 | 97.94 | 32.69 | 74.35 | 63.48 | 67.11 |
| Text-IF | 71.75 | 80.91 | **45.67** | 41.17 | 40.73 | 15.16 | 49.23 | **98.02** | 40.43 | **76.03** | 58.53 | 68.25 |
| GIFNet | 69.83 | 79.64 | 40.25 | **47.25** | 32.94 | 31.79 | 50.28 | 97.85 | 35.98 | 73.19 | 61.30 | 67.08 |
| TEDFusion | **75.35** | **81.98** | **54.62** | **48.11** | **43.01** | **41.13** | **57.37** | **98.07** | **42.08** | **75.58** | 62.85 | **69.64** |

## 4.2. Comparative Results Analysis

**Qualitative Comparisons.** The visual comparisons in Fig. 3 demonstrate the superior performance of our method. Overall, the proposed approach excels in two main aspects. First, it achieves precise semantic alignment through hyperbolic manifold learning in the Poincaré ball, where image features are pulled toward text-based centroids to enhance attribute fidelity. This is evident in TNO, where house edges are distinctly preserved against the sky due to optimized gradient representation, and in RoadScene, where the texture complexity of trees and mountains is enhanced, yielding sharper contours. Second, the method exhibits robust hierarchical modelling that adapts to challenging conditions. For instance, in MSRS, wheels in low-light scenarios are brightened with improved contrast, ensuring clarity, while

in LLVIP, logos on the ground are emphasized through saturation and gradient adjustments, reinforcing semantic accuracy. The negative curvature of the hyperbolic space enables natural integration of multi-modal features, consistently advancing performance across diverse environments.

**Quantitative Comparisons.** The quantitative results strongly validate the advantages of our hyperbolic manifold modelling. As shown in Table 1, the superior performance across EN, SD, SF, AG, and SCD metrics indicates that our fusion results comprehensively preserve the informational entropy and gradient structures of the source images. Specifically, the high scores in AG and SF underscore an enhanced representation of texture and sharpness, which directly stems from our method's ability to hierarchically organize and refine gradient information within the nega-

tively curved Poincaré ball. Furthermore, the notably high SD value is a direct result of the enhanced contrast achieved through hyperbolic alignment. This process prioritizes semantically critical features from both modalities, yielding high-definition fused images that excel in perceptual quality.

### 4.3. Ablation Study

**Cross-Modal Training and Fusion Mechanism.**

To validate the effectiveness of our cross-modal training strategy, we conduct two ablation experiments on the infrared and visible image fusion task. In Exp. I, we completely remove the text guidance during both training and testing phases, forcing the model to rely solely on raw image features, which results in significant performance degradation. In Exp. II, the model is trained using only text features but tested on attribute features alone, which severs the semantic guidance and leads to the loss of fine-grained image details. Beyond the training strategy, we ablate the proposed ATGFM in Exp. III. This removal eliminates adaptive attribute modulation and semantic-guided enhancement, resulting in degraded image quality. Integrating insights from all three variants, our full method achieves optimal performance by incorporating text during training and maintaining the ATGFM-driven semantic guidance, resulting in superior fusion quality across both quantitative metrics (Table 2) and visual quality (Fig. 4), particularly in preserving thermal radiation information and texture details.

**Hyperbolic Modelling.**

To validate the efficacy of hyperbolic modelling, we conduct two ablation experiments. In Exp. IV, cross-modal association is performed in Euclidean space instead of the hyperbolic Poincaré ball, removing the exponential mapping and Möbius operations. In Exp. V, we replace the adaptive hyperbolic loss function with standard Euclidean distance measurement. The Euclidean association experiment fails to capture hierarchical semantic relationships, leading to ambiguous fusion outputs with reduced structural coherence, while removing the hyperbolic loss further exacerbates the misalignment, particularly in semantically critical regions. This is largely because the hyperbolic loss alters the direction of gradient propagation: when computing the gradient of the Poincaré distance, features near the ball center receive stronger backpropagation signals, prompting the model to prioritize optimizing regions corresponding to core semantics of the text (*e.g.* key objects like "man" or "lamp"). These results, as demonstrated in Table 2 and Fig. 5, collectively underscore the importance of curvature-aware geometric reasoning for robust cross-modal fusion.

**Different Curvature Settings.** To further validate the impact of curvature, we configure the Poincaré ball with differ-

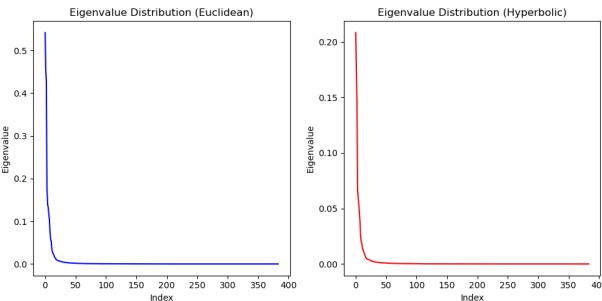

*Figure 8.* Comparison of eigenvalue distributions between Euclidean and Hyperbolic feature spaces.

*Table 4.* Quantitative distribution of the top 10 eigenvalues. The hyperbolic space exhibits a significantly more gradual spectral decay.

| Rank | Euclidean Eigenvalues | Hyperbolic Eigenvalues |
|------|------------------------|-------------------------|
| 1 | 0.5417 | 0.2083 |
| 2 | 0.4532 | 0.1752 |
| 3 | 0.4283 | 0.1455 |
| 4 | 0.1765 | 0.0669 |
| 5 | 0.1408 | 0.0604 |
| 6 | 0.1323 | 0.0536 |
| 7 | 0.1177 | 0.0468 |
| 8 | 0.1027 | 0.0380 |
| 9 | 0.0754 | 0.0251 |
| 10 | 0.0578 | 0.0199 |

ent curvature parameters in Exp. VI ($c = 0.5$) and Exp. VII ($c = 0.7$), where smaller curvature values correspond to flatter geometric spaces with weaker hierarchical modelling capacity. As shown in Table 2, reducing the curvature leads to performance degradation, particularly in SF and SD, which measure spatial frequency and image contrast. This manifests visually as blurred edges and loss of salient details, as the flatter hyperbolic space fails to preserve the hierarchical semantic structure essential for distinguishing foreground from background. Our full method in Exp. VIII ($c = 1.0$) achieves optimal performance by leveraging sufficient curvature to encode semantic hierarchy, thereby maintaining sharp boundaries and thermal details.

**Eigenvalue Distribution Analysis.** To quantitatively validate that the inherent flatness of Euclidean spaces induces representation collapse, we perform an analysis of the covariance matrices of features extracted from both Euclidean-based baselines and our TEDFusion. As illustrated in Fig. 8 and Table 4, Euclidean eigenvalues exhibit a sharp, heavy-headed decay. Specifically, the top three components account for a disproportionate majority of the total variance while subsequent values diminish rapidly. This indicates a "feature flattening" phenomenon where information is compressed into a low-dimensional subspace, erasing fine-grained details. In contrast, by mapping features onto the Poincaré ball, TEDFusion demonstrates a notably more bal-

anced eigenvalue distribution with an attenuated decay rate. This suggests that hyperbolic geometry preserves a higher intrinsic dimensionality, distributing information more equitably across the embedding space to capture complex, multi-scale semantic hierarchies.

### 4.4. Downstream Experiments

**Object Detection.** We evaluate TEDFusion on an object detection task using the M3FD dataset. With a 7:3 train-test split and YOLOv7 (Wang et al., 2023) as the evaluator, our approach achieves the highest mAP, as summarized in Table 3. This quantitative superiority is further supported by visual comparisons in Fig. 6, where TEDFusion demonstrates a superior ability to accurately localize diverse objects within complex scenes, whereas competing methods suffer from either missed detections or false positives. This precision, attributable to the rich informational integrity maintained by TEDFusion, validates the efficacy of our semantic adaptive strategy (see Appendix B for detection details).

**Semantic Segmentation.** The semantic segmentation performance is evaluated using DeepLabV3+ network (Chen et al., 2018). The results in Table 3 show that our method achieves competitive performance in the four-class task, as reflected in its overall accuracy and mIoU. This effectiveness is supported by the qualitative evidence in Fig. 7, where our approach produces more precise object boundaries and better spatial coherence. These findings indicate that our fusion technique contributes to reliable pixel-wise accuracy and structurally consistent segmentation (see also Appendix B).

## 5. Conclusion

We present TEDFusion, a novel adaptive framework for infrared and visible image fusion. During training, semantic text guidance helps the model learn effective fusion strategies. At inference, it operates in a fully text-free manner, dynamically adjusting the fusion process based solely on visual inputs to enhance critical scene attributes. By embedding features in a hyperbolic manifold, our method achieves a more native and hierarchical alignment across modalities compared to Euclidean-based approaches, leading to perceptually consistent fusion results. Extensive experiments show that TEDFusion outperforms existing methods across multiple benchmarks.

## Acknowledgements

This work was supported by the National Key Research and Development Program of China (2023YFE0116300), the National Natural Science Foundation of China (62202205, 62576152, 62332008), the Basic Research Program of Jiangsu (BK20250104), the Fundamental Research Funds for the Central Universities (JUSRP202504007).

## Impact Statement

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

## A. More Detail in Computation of GLCM.

In the implementation, structural attributes are encoded through multi-scale GLCM (Haralick & Dinstein, 2007; Singh et al., 2017) where the input RGB tensor is first converted to grayscale, then the grayscale image is spatially summed and normalized to the range [0, 255] as an 8-bit unsigned integer representation. The GLCM is computed with four distance offsets $d \in \{2, 4, 8, 16\}$ pixels and four angular directions $\theta \in \{0, \frac{\pi}{4}, \frac{\pi}{2}, \frac{3\pi}{4}\}$ across all 256 gray levels, with symmetric and normalized options enabled to ensure the matrix $P(i, j)$ represents joint probability distributions.

From this co-occurrence matrix, energy (also known as Angular Second Moment) is extracted, which measures the uniformity or orderliness of the texture by computing the sum of squared elements in the GLCM:

$$E = \sum_{i=0}^{255} \sum_{j=0}^{255} P(i, j)^2 . \tag{10}$$

Homogeneity (also called Inverse Difference Moment) quantifies the closeness of the distribution of GLCM elements to the diagonal and reflects local texture smoothness:

$$H = \sum_{i=0}^{255} \sum_{j=0}^{255} \frac{P(i, j)}{1 + |i - j|} . \tag{11}$$

The final texture complexity metric $T_c$ is constructed by linearly combining these two measures with modality-specific weights, where the average values across all scales and orientations are used:

$$T_c^{IR} = 0.7E + 0.3H, \tag{12}$$

$$T_c^{VI} = 0.6E + 0.4H . \tag{13}$$

The weight allocation reflects the distinct textural characteristics of each modality: infrared images typically exhibit more uniform and structured patterns due to thermal radiation properties, thus assigning higher weight (0.7) to energy emphasizes texture orderliness; conversely, visible images contain richer local variations and edge details from reflected light, warranting increased weight (0.4) on homogeneity to better capture fine-grained smoothness transitions.

Additionally, spatial contrast $C_s$ is independently calculated as the mean standard deviation across color channels, where $\mu$ is the channel-wise mean and $H, W$ are spatial dimensions:

$$C_s = \text{mean} \left( \sqrt{\frac{1}{HW} \sum_{h,w} (I_{h,w} - \mu)^2} \right) . \tag{14}$$

This complementary measure provides a direct assessment of intensity dispersion that, combined with the texture complexity metric, offers a comprehensive characterization of both structural patterns and statistical variations within the image.

## B. Specific Configuration of Downstream Tasks.

In the detection task, we conduct experiments using YOLOv7 on the M3FD dataset, splitting the data with a 7:3 ratio for training and testing across diverse scenarios including rainy roads, foggy fields, nighttime urban streets, and daytime natural scenes. The model is trained for 200 epochs with an input image size of $640 \times 640$ pixels and a batch size of 8, utilizing SGD optimizer with an initial learning rate of 0.01 and momentum of 0.937. The training employs a cosine annealing learning rate scheduler that gradually reduces the learning rate to 0.1% of its initial value, along with warmup epochs of 3.0 to stabilize early training. We evaluate model performance using mAP@0.5 metric across six object categories: Person, Car, Bus, Lamp, Motor, and Truck.

We evaluate the semantic segmentation performance using DeepLabV3+ on the MSRS dataset, employing images across diverse environmental conditions including nighttime roads, daytime streets, factories, residential areas, and construction sites. The training process spans 100 epochs with an input resolution of $480 \times 640$ pixels and batch size of 8, utilizing SGD optimizer with an initial learning rate of 0.007 and momentum of 0.9. We set the downsample factor to 16 to balance computational efficiency and segmentation accuracy, and apply weight decay of 0.0001 to prevent overfitting. The dataset

is split with a 9:1 ratio for training and validation, and we evaluate model performance using mean Intersection over Union (mIoU) and class-wise average precision for four object categories: background, person, car and bike. The model is initialized with pretrained DeepLabV3+ weights and optimized using standard cross-entropy loss.

