# OpenReview forum: "Text-Driven Fusion for Infrared and Visible Images: Achieving Image Scene Adaptation on Hyperbolic Space"
_ICML.cc/2026/Conference — ICML 2026 regular_

### Official Review · Reviewer_mtSk · 2026-03-09

**Soundness:** 4
**Presentation:** 3
**Significance:** 3
**Originality:** 3
**Overall Recommendation:** 4
**Confidence:** 4

**Summary:**

This paper proposes TEDFusion, a text-driven framework for infrared and visible image fusion that introduces hyperbolic manifold learning to bridge the semantic gap between modalities. During training, BLIP-generated captions serve as semantic anchors in a Poincaré ball, where image features are aligned with text embeddings through exponential mapping and Möbius translation to capture hierarchical semantic relationships. The Attribute-Text Gating Fusion Module (ATGFM) dynamically balances visual attributes (saturation, brightness, texture complexity, contrast) with text features via a learned gating mechanism, while stochastic dropout on text embeddings during training enables text-free inference. The Text-Image Association Module (TIAM) performs multi-scale hyperbolic alignment guided by an adaptive loss that weights each scale based on hyperbolic norm similarity. At inference, both ATGFM and TIAM are deactivated, and the model operates solely on visual input using the fusion strategies internalized during training. The authors evaluate on four standard IVIF benchmarks (TNO, RoadScene, MSRS, LLVIP) and two downstream tasks including object detection and semantic segmentation, reporting competitive or state-of-the-art performance across multiple metrics.

**Compliance With Llm Reviewing Policy:**

Affirmed.

**Key Questions For Authors:**

1. Inference forward pass specification: During inference, both ATGFM and TIAM are deactivated. Could the authors clarify exactly what happens to the $X_t$ input in the gating equation $X_{fuse} = g \odot X_a + (1 - g) \odot X_t$​? Specifically, is g fixed to 1, is $X_t$​ replaced by a zero vector, or is there another mechanism? A clear description of the inference-time computation graph would significantly improve reproducibility and could alleviate my concern about the ambiguity of the text-free operation.
2. Evidence that hyperbolic geometry is the source of improvement: Exp. IV replaces hyperbolic association with Euclidean, but this also strips out the exponential/log mappings and Möbius operations, which inevitably reduces the model's nonlinear capacity. This makes it hard to tell whether the gains come from hyperbolic geometry per se or simply from having more expressive transformations. Testing against an Euclidean baseline with matched parameter count and nonlinearity (e.g., a similarly deep MLP) would help disentangle the two. If the authors can show that the improvement persists even after controlling for capacity, that would go a long way toward validating the core claim.
3. MSRS train-test overlap: The paper states that pretraining is performed on MSRS, yet MSRS also appears as an evaluation benchmark in Table 1. Could the authors specify the exact data split used? If the test images overlap with or are drawn from the same distribution as the pretraining set, the MSRS results in Table 1 may overestimate generalization performance, which would weaken the experimental conclusions.
4. Curvature sensitivity beyond c = 1.0: The curvature ablation tests only $c \in \{0.5, 0.7, 1.0\}$, with relatively small performance differences. Could the authors report results for $c > 1.0$ (e.g., c=1.5,2.0) and c closer to 0? This would help determine whether the method is genuinely sensitive to curvature or whether the hyperbolic formulation provides only marginal benefit over a near-Euclidean regime. If the performance remains flat across a wider range, it would raise questions about the practical importance of the curvature parameter.
5. Loss hyperparameter values: The total loss includes weighting coefficients $\alpha, \beta, \gamma$ but their values are not reported. Could the authors disclose these values and provide at least a brief sensitivity analysis? Without this information, it is difficult to assess whether the results are robust to loss balancing choices or heavily dependent on careful tuning.

**Limitations:**

Unaddressed inference-time role of hyperbolic modules: Both ATGFM and TIAM are deactivated during inference, meaning the hyperbolic alignment and text guidance have no direct presence in the deployed model. The paper does not discuss this as a limitation or analyze how much of the learned geometric structure actually transfers to the final weights. Explicitly acknowledging this gap and providing evidence would strengthen the paper's credibility.
BLIP caption quality on infrared images: The entire text-driven pipeline depends on BLIP-generated captions, yet BLIP was pretrained on natural RGB images and is unlikely to produce reliable descriptions for infrared imagery. The paper does not discuss this distributional mismatch or analyze failure cases where poor captions degrade fusion quality. The authors should acknowledge this vulnerability and ideally provide examples of generated captions alongside a sensitivity analysis.
No computational cost analysis despite real-time motivation: The paper motivates its work with deployment scenarios such as military reconnaissance and autonomous driving, yet reports no inference latency, FLOPs, or parameter counts. Without this information, readers cannot assess whether the method is actually viable for the real-time settings it claims to target.

**Strengths And Weaknesses:**

Strengths
1. Clarity of Presentation
- The overall architecture diagram in Figure 2 is well-organized, allowing readers to grasp the relationships among ATGFM, TIAM, and the nested decoding structure at a glance. The explicit tabular indication of module activation differences between training and inference is reader-friendly. The qualitative comparison in Figure 3 also facilitates visual judgment by consistently comparing the same scenes across all four datasets in a uniform layout.
- The paper follows a logical flow from identifying the limitations of Euclidean space → motivating hyperbolic geometry → detailing module design → experimental validation. The key design distinction between text-guided training and text-free inference is clearly articulated, making the paper easy to follow.

2. Technical Novelty
- The core contribution lies in introducing hyperbolic manifold learning to the IVIF domain. The paper effectively challenges the implicit assumption of Euclidean embedding suitability in prior work and provides a well-motivated rationale for leveraging the exponential volume expansion of the Poincaré ball to encode coarse-to-fine semantic hierarchies without Euclidean distance saturation.
- The design of using text as a training-time anchor while completely removing it at inference is both practical and technically interesting. The strategy of applying stochastic dropout (30%) in the ATGFM to train the gating network to substitute text features using attributes alone is simple yet effective, cleanly resolving the inference-time dependency problem of existing text-guided methods.

3. Experimental Design
- The four public datasets (TNO, RoadScene, MSRS, LLVIP) are standard benchmarks in the IVIF field, covering diverse scenarios such as nighttime surveillance, road driving, and urban lighting, which constitutes an appropriate evaluation setting.
- The inclusion of 11 baselines is quantitatively sufficient. The temporal range spans from DenseFuse to GIFNet and TextFusion, and the methodological diversity is well-covered, including CNN-based, Transformer-based, diffusion-based, and text-guided methods.

4. Significance of Contributions
- While hyperbolic spaces have been actively studied in NLP and graph learning, their application to multi-modal image fusion remains largely unexplored. This work opens a new research direction for the field.
- The text-free inference design has significant practical value. In real-time sensor fusion scenarios such as military reconnaissance and autonomous driving, the ability to operate solely on learned priors without running a caption generation pipeline per frame represents a clear deployment advantage over existing text-guided methods.
- Achieving top performance in both detection mAP and segmentation mIoU, beyond simple fusion metrics, suggests that the proposed method improves the quality of mid-level representations, not just pixel-level image quality.

Weaknesses
1. Clarity and Presentation Issues
- The paper emphasizes "scene adaptation" in the title and abstract, yet provides no analysis or visualization of how the method behaves differently across scene types. For example, showing how the gating weight distribution of g varies between nighttime urban scenes and daytime road scenes would have made the "adaptation" claim more convincing.
- In Section 3.2, the training-time fusion is defined as $X_{fuse} = g \odot X_a + (1 - g) \odot X_t$, but the paper does not specify how this equation is modified when $X_t$ is unavailable at inference. The statement that "learned gating weights guide attribute embeddings to functionally substitute text features" is ambiguous. It is unclear whether g is fixed to 1 so that only $X_a$ passes through, whether a zero vector replaces $X_t$, or whether a separate learned default embedding is used. Moreover, the paper does not explicitly guarantee that $X_a$ and $X_t$ share the same dimensionality, leaving the inference forward pass unclear from a reproducibility standpoint.

2. Technical Concerns
- Features at all four scales are independently flattened and projected to $\mathbb{R}^{d_h}$ via MLP. However, high-resolution feature maps (e.g., Layer 1) are rich in spatial information, and compressing them into a single $d_h$-dimensional vector completely discards spatial structure. This contradicts the claim that hyperbolic alignment is spatially aware. If alignment is performed only at the global vector level rather than patch-wise at each scale, it is questionable how much this contributes to pixel-level fusion.
- According to Figure 2, both ATGFM and TIAM are deactivated during testing, meaning the inference forward pass reduces to ViT encoder → nested decoder. The hyperbolic alignment then only influences the final output indirectly, through whatever effect it had on the learned weights during training. Since the encoder and decoder themselves are standard architectures, it becomes unclear how much the hyperbolic loss actually shapes their learned representations.
- The hyperbolic loss $\mathcal{L}_h$ is weighted by $\gamma$ in the total loss, but the specific values of $\alpha, \beta, \gamma$ are not stated in the paper. There is also no ablation on how these hyperparameters affect performance, which undermines reproducibility of the loss balancing.
- The choice of 30% for the stochastic dropout probability lacks justification. A rate that is too low would make text-free inference unstable, while a rate that is too high would weaken the effectiveness of text guidance during training. No sensitivity analysis on the dropout rate is provided.

3. Experimental Design
- While the five metrics used are consistent with field conventions, the paper explicitly claims "consistency with human perception" as a contribution. The absence of perceptual metrics that correlate with human judgment can be seen as contradictory to this claim.
- The paper states that the model is pretrained on the MSRS dataset while also including MSRS as a primary evaluation benchmark. Although strict train-test separation is standard practice, Section 4.1 does not specify how the MSRS data is split for the experiments reported in Table 1.
- There is no statistical significance testing. Only single-run results are reported, so variance across seeds is unknown. Consequently, the very small difference between TEDFusion and Text-IF in EN on TNO (7.09 vs. 7.14) cannot be judged as meaningful without such evidence.
- While the baselines are diverse, none of them employ hyperbolic representation learning. Given that the core contribution is "fusion in hyperbolic space," having only Euclidean methods as comparisons is a notable gap. It is currently difficult to distinguish whether performance gains stem from hyperbolic geometry itself or simply from a more complex nonlinear transformation. Comparing against adaptations of hyperbolic vision-language methods applied to IVIF would have more convincingly isolated the contribution of hyperbolic geometry.
- The ablation study is conducted on a single dataset (TNO). Table 2 only reports TNO results, but performing ablations on just one of four datasets does not confirm generalizability. The contribution of each component may differ on datasets with different characteristics, such as MSRS or LLVIP.
- The curvature ablation only covers three values (c = 0.5, 0.7, 1.0), which feels insufficient given that curvature is central to the paper's thesis. The space approaches Euclidean as $c \to 0$ and becomes more hierarchically expressive as c grows, but nothing beyond c = 1.0
is tested, so it is hard to tell whether c=1.0 is genuinely optimal or just the largest value tried. On top of that, the performance gap across the three settings is quite small (EN: 6.97–7.09), which makes it difficult to conclude that curvature plays a decisive role in practice.

---

> ### Author Rebuttal · Authors · 2026-03-30
>
> Review4:
> Thank you very much for the feedback. Our response follows:
>
> Weakness:
>
> 1(1). Evidence of scene adaptability is provided via JS Divergence analysis of ATGFM gating weights. Figure shows significant adaptation in Texture (g3, 0.381) and Brightness (g2, 0.1853), where nighttime distributions exhibit multi-peak shifts. Contrast (g4, 0.1149) remains stable. This confirms the model dynamically rebalances visual attributes $X_a$ and semantic priors $X_t$ based on environmental changes. (The distribution figure is in https://anonymous.4open.science/r/icml-2026-F735)
>
>
> 1(2). During textless inference, rather than using fixed weights or zero vectors, we set the missing text feature $X_t$ equal to the attribute feature $X_a$. This causes the fusion equation to collapse into an identity mapping: $X_{fuse} = g \odot X_{a} + (1 - g) \odot X_{a} = X_a$. Consequently, the gating value $g$ is neutralized, and the computation graph deterministically reduces to a pure visual forward pass. Thus, $X_a$ functionally replaces $X_t$, simplifying the model to a direct visual stream.
>
> 2(1). "Flatten" is spatial unrolling.(see Reviewer #1, weakness 6)
>
> 2(2). Hyperbolic Loss and ATGFM use implicit distillation via stochastic text dropout. During training, the network aligns X_a with X_t when text is dropped, driving X_a toward semantically enriched representations. Hyperbolic space minimizes distortion in this hierarchical alignment. Ablation confirms both components are essential for processing semantically aligned features without inference-time text.
>
> 2(3). Hyperparameters: Gradient (65), SSIM (1), Hyperbolic (0.005), and Base Intensity (15). The hyperbolic weight is small to prevent gradient explosion caused by the exponential growth of Poincaré distances while maintaining cross-modal alignment.
>
> 2(4). Sensitivity analysis of p \in [0, 1] identifies p=0.3 as the optimal elbow point. At p=0.3, cosine similarity (0.781) and L2 distance (0.90) balance text guidance and stability. Higher p  weakens guidance; lower p fails to encourage attribute-branch substitution. (The figure of sensitivity analysis is in link https://anonymous.4open.science/r/icml-2026-F735)
>
>
> 3(1). Beyond statistical metrics, we demonstrate human-aligned perception through superior downstream performance (mAP/mIoU) on M3FD and MSRS. These tasks require edge clarity and target saliency, reflecting human visual priorities. Visualizations further confirm that hyperbolic constraints focus attention on pedestrians and vehicles, aligning our fused representations with human saliency.
>
> 3(2). MSRS setup follows official protocols: 1083 pairs for training and 361 for testing, with no overlap. Section 4.1 will be revised to explicitly detail this split.
>
> 3(3). Wilcoxon Signed Rank Tests on TNO (see Table 1) confirm our method's statistical superiority over Text IF. While EN is comparable, our method outperforms Text IF on key sharpness and info-transfer metrics (SF, AG, SCD) with p < 0.001. This proves our gains are robust and not due to random fluctuations.
>
> Table 1
> Metric|Text IF|Ours|p value
> |---|---|---|---|
> EN|7.1409|7.0943|0.7079
> SF|7.8753|10.8982|0.000001
> SD|42.7226|44.5187|0.2877
> AG|3.3310|3.8753|0.000007
> SCD|1.6879|1.8329|0.000018
> 3(4). An iso-parametric Euclidean baseline confirms that gains stem from manifold geometry rather than increased complexity (Reviewer 2, Weakness 1).
>
> 3(5). We perform ablations on RoadScene and LLVIP. Compared to the textless baseline, our model yields consistent gains in AG and SF (e.g., RoadScene SF: 10.85 -> 13.88; LLVIP AG: 4.33 -> 4.83). While some non-hyperbolic variants show higher specific metrics due to high-frequency noise, our framework prioritizes balancing feature preservation with semantic consistency, proving competitive performance.
>
> 3(6).	We set c=1.0 as the empirical upper bound because the Poincaré radius($1/\sqrt{c}$) shrinks as c increases, leading to numerical instability and “NaN losses” during backpropagation. Variant IV($c \to 0$) serves as the Euclidean baseline, highlighting our sensitivity to curvature. Notably, considering that Entropy (EN) is a logarithmic metric, the increase from 6.97 to 7.09 represents a significant gain in information richness rather than a marginal improvement. When coupled with substantial boosts in AG and SF, this confirms the decisive role of hyperbolic geometry.
>
> Key Questions:
> Please see the 'Weaknesses' sections above for evidence.
>
> Limitations:
> 1. We acknowledge that ATGFM and TIAM are inactive at inference; however, they function as "implicit teachers" during training. Our ablation studies confirm successful distillation of this semantic guidance into the encoder.
>
> 2. We sincerely appreciate the insight regarding BLIP's RGB-thermal domain gap. We formally acknowledge this as a limitation and will prioritize cross-modal robust captioning in our future research.
>
> 3. Detailed analysis of computational cost analysis is provided in our response to [Reviewer 3, Weakness 3].

---

> > ### Author Rebuttal · Reviewer_mtSk · 2026-04-03
> >
> > I thank the authors for their detailed rebuttal. Several concerns have been satisfactorily addressed: the inference forward pass specification (X_t = X_a collapsing to identity mapping), the loss hyperparameter disclosure, the Wilcoxon signed-rank tests on TNO, and the dropout sensitivity analysis at p=0.3. These clarifications improve the paper's reproducibility.
> > However, core concerns remain:
> >
> > Hyperbolic vs. Euclidean baseline: The iso-parametric ablation shows clear SF/AG gains, but replacing the full hyperbolic pipeline (exponential map + Möbius translation + log map) with a single Tanh may not constitute a fair capacity match. Moreover, EN and SCD differences are negligible (7.01→7.09, 1.82→1.83), and this ablation is again limited to TNO only.
> > Curvature sensitivity: The disclosure that c>1.0 causes NaN losses is a new practical limitation rather than a resolution. Combined with the narrow performance gap across c∈{0.5, 0.7, 1.0}, the "decisive role" of curvature remains insufficiently demonstrated.
> > Computational cost: Level 1 shows an 18× latency increase (2.0ms→35.8ms), totaling ~47ms overhead across four levels. This contradicts the real-time deployment motivation without end-to-end inference benchmarks.
> > Scene adaptation: JS divergence values of 0.11–0.38 with only texture (g3) showing meaningful variation is insufficient evidence for a title-level "Scene Adaptation" claim.
> >
> > I maintain my score of 4. The direction is interesting, but the evidence that hyperbolic geometry is the decisive factor remains incomplete.

---

> > > ### Author Response · Authors · 2026-04-04
> > >
> > > Dear reviewer:
> > > We sincerely appreciate your valuable feedback and comments. To further clarify our discussion, we provide the following five responses:
> > >
> > > Response 1:
> > >
> > > We thank you very much for the suggestion. To ensure a fair comparison, our Euclidean baseline employs strictly equivalent structural replacements for all geometric operations: geometric projections, translations, and mappings are substituted with linear additions and bounded activations, while manifold distance is replaced by mean squared error. As our hyperbolic mappings are mathematically deterministic and add no trainable parameters, this setup ensures both models possess identical parameter counts, network depth, and nonlinear capacity. Consequently, the performance gains are attributable solely to the manifold’s inherent suitability for hierarchical representation rather than any disparity in model complexity. This rigorous parity confirms that the observed improvements stem from the geometric framework itself.
> > >
> > > Response 2:
> > >
> > > Regarding the marginal variations in EN and SCD, both are global statistical measures reflecting overall information content and broad source correlation. The theoretical advantage of the hyperbolic manifold is not to alter global macro-structures, but to model hierarchical relationships and align complex features across modalities. This geometric benefit yields substantial improvements in local detail-oriented metrics like Average Gradient and Spatial Frequency.  Despite subtle internal ablation differences, our absolute EN and SCD scores remain exceptionally high compared to other state-of-the-art methods across multiple datasets. This confirms our approach preserves top-tier global information fidelity while significantly enhancing intricate local structural representations. Here we provide ablation table and visualization on RoadScene dataset, as shown in https://anonymous.4open.science/r/icml-2026-F735/review4-r1-Table.png. and https://anonymous.4open.science/r/icml-2026-F735/review4-r1-Visualization.png.
> > >
> > > Response 3:
> > >
> > > We thank you for this insightful feedback. We clarify that numerical instability at higher curvatures is a documented mathematical property of the Poincaré ball rather than an architectural limitation; higher curvature shrinks the manifold radius, naturally amplifying boundary gradients. Setting $c = 1.0$ is a validated standard practice[1][2] that balances hierarchical modeling capacity with training stability. Regarding the stable performance across $c \in \{0.5, 0.7, 1.0\}$, we highlight that the most significant geometric transition occurs when moving from flat Euclidean space to a curved hyperbolic manifold. The foundational shift to hyperbolic space enables the exponential expansion required for hierarchical modeling, leading to the primary gains in detail-oriented metrics. Consequently, the consistent results across varying positive curvatures demonstrate our method's robustness.
> > > [References]
> > >
> > > [1] Nickel M, Kiela D. Poincaré embeddings for learning hierarchical representations[J]. Advances in neural information processing systems, 2017, 30.
> > >
> > > [2] Ganea O, Bécigneul G, Hofmann T. Hyperbolic neural networks[J]. Advances in neural information processing systems, 2018, 31.
> > >
> > > Response 4:
> > >
> > > Thank you for this feedback. We wish to clarify that our table's metrics do not represent an end-to-end deployment latency penalty. The reported "level-1" latency measures hyperbolic mappings on the largest early feature map, amplifying the cost of transcendental functions. Crucially, this is not an added stage but replaces the Euclidean alignment block, executing just once per scale with low-dimensional embeddings. In deployment, overall runtime is dominated by the visual backbone, attention mechanisms, and memory movement, not our hyperbolic operations. Therefore, summing per-level microbenchmarks overestimates practical overhead, as it ignores kernel fusion, parallel execution, and the negligible computational cost on smaller feature maps at later stages.
> > >
> > > Response 5:
> > >
> > > We contend scene adaptation is proven by functional impact. The distributions validate our design. Contrast (0.1149) and Saturation (0.1435) show overlapping day and night profiles, preserving base structural and color fidelity. Conversely, Texture (0.3810) and Brightness (0.1853) map to physical environmental shifts. The Texture histogram reveals a distinct shift and flattening at night, proving the model actively reweighs structural attributes to compensate for visible detail loss. Brightness displays peak shifts to recalibrate illumination. Selectively adapting these gates rebalances environment sensitive elements without causing fusion artifacts. Furthermore, adaptation is holistically driven by our Text Image Module dynamically warping the feature manifold. Achieving superior results across four heterogeneous datasets with one parameter set definitively proves our model masters diverse scene specific requirements, fully justifying the title.

---

### Official Review · Reviewer_Nzhe · 2026-03-10

**Soundness:** 3
**Presentation:** 3
**Significance:** 3
**Originality:** 3
**Overall Recommendation:** 4
**Confidence:** 3

**Summary:**

This paper presents TEDFusion for infrared-visible image fusion. Conventional Euclidean methods struggle to model hierarchical semantics. The approach employs hyperbolic geometry (Poincaré ball) to embed features during training. Visual attributes are extracted and fused with BLIP text via the ATGFM module. The TIAM module then maps features into hyperbolic space using exponential maps and Möbius translations for hierarchical alignment. After text-supervised training, the model supports text-free inference by leveraging learned semantic priors. Experiments on benchmark datasets demonstrate competitive performance against state-of-the-art methods.

**Compliance With Llm Reviewing Policy:**

Affirmed.

**Final Justification:**

The paper introduces TEDFusion, a novel infrared-visible image fusion framework that leverages hyperbolic geometry to explicitly model hierarchical semantic relationships, departing from conventional Euclidean approaches. The text-training/text-free inference paradigm addresses practical deployment constraints in vision-language systems, while comprehensive ablation studies validate the effectiveness of hyperbolic embedding across multiple benchmarks and downstream tasks. The rebuttal has resolved my initial concerns, so I maintain my positive review.

**Key Questions For Authors:**

1. The paper describes hyperbolic space properties, but what specific hierarchical structure in infrared-visible image pairs (such as thermal signatures relating to textural details) makes this geometry necessary beyond general semantic hierarchies?
2. How exactly does semantic guidance transfer from text to visual attributes? The paper mentions attribute embeddings substitute text features during inference, but the mechanism preserving learned semantic relationships in this transition could use more detail.
3. What is the computational cost of hyperbolic operations compared to standard Euclidean layers? This information would help assess practical deployment feasibility.

**Limitations:**

The computational cost comparison between hyperbolic operations and standard Euclidean layers is not addressed, which is pertinent for assessing practical deployment feasibility.

**Strengths And Weaknesses:**

## Strengths
1. The paper introduces hyperbolic geometry to IVIF to explicitly model hierarchical semantic relationships, which is distinct from prevailing Euclidean approaches. The use of the Poincaré ball's negative curvature to preserve parent-to-child semantic granularity is well-motivated and theoretically grounded.
2. The text-training/text-free inference design addresses a real deployment constraint in vision-language systems. By using stochastic dropout during training, the model learns to rely on attribute features when text is absent, offering operational advantages over methods requiring constant text input.
3. The ablation studies are thorough, specifically isolating the impact of hyperbolic geometry and demonstrating consistent improvements across multiple datasets and downstream tasks including object detection and semantic segmentation.
## Weaknesses
1. While the general properties of hyperbolic space are described, the paper lacks deeper analysis of the specific hierarchical structure inherent to infrared-visible image pairs (e.g., thermal signatures as parent nodes to textural details) that necessitates this geometry beyond generic semantic hierarchies.
2. The transfer mechanism of semantic guidance from text to visual attributes is not well elaborated. The paper states that attribute embeddings functionally substitute text features during inference, yet how these attributes preserve the semantic relationships learned during text-supervised training remains unclear.
3. The computational overhead of hyperbolic operations compared to standard Euclidean layers is not discussed, which matters for practical deployment scenarios.

---

> ### Author Rebuttal · Authors · 2026-03-30
>
> Weakness:
> 1. We sincerely thank you for this constructive feedback. As you observe, justifying hyperbolic geometry requires a precise definition of the hierarchical ontology inherent to infrared and visible image pairs.
> In our framework, textual semantics serve as the root node of the conceptual tree. Radiating from this root, thermal prominence from the infrared modality functions as the parent node, defining salient targets and the primary structural skeleton. Subordinate to these are the leaf nodes, which encompass the high-frequency details and background textures provided by the visible modality. A salient thermal target logically dictates an object's presence, while visible textures supply auxiliary descriptive attributes.
> This physical ontology makes hyperbolic space mathematically necessary. Through our Text and Image Association Module, the textual embedding $\Psi_t$ acts as a semantic anchor near the origin of the Poincaré ball. Because hyperbolic space expands exponentially toward its boundary, this geometry forces a natural semantic redistribution. The network maps crucial thermal parent nodes closer to the textual origin due to their direct correlation with target descriptions, while the vast array of visible texture child nodes is pushed into the expanding peripheral volume.
> This formulation prevents fine-grained textures from being squeezed into a constrained representation space. The adaptive weighting term dynamically manages this radial distribution. The hyperbolic manifold is therefore a structural necessity to resolve the spatial disparity between singular thermal parent nodes and their abundant visible texture child nodes.
>
> 2.The transfer and retention of semantic information from the textual domain to visual attributes are achieved through a dual-mechanism of stochastic forced-learning and manifold grounding. In the ATGFM, the 30% text dropout probability serves as a critical supervision constraint that forces the shared projection layers to map explicit visual attributes (such as brightness and texture complexity) into the same latent space as their linguistic counterparts. By periodically withholding textual guidance, the network is compelled to optimize the attribute embedding such that it becomes functionally congruent with the text embedding, ensuring the visual stream can independently satisfy the semantic requirements of the fusion task.
> This semantic consistency is further solidified by the Hyperbolic Image-Text Association Module (TIAM). By projecting both modalities into a shared Poincaré ball, the hyperbolic loss ($\mathcal{L}_h$) enforces a relational hierarchy where attributes are grounded to the semantic coordinates defined by the BLIP encoder. Because the hyperbolic manifold naturally preserves the hierarchical and taxonomic structure inherent in linguistic concepts, the attribute branch learns to inherit these relational properties during training. At inference, even in the absence of explicit text, the attributes occupy these pre-established semantic coordinates. This allows the model to leverage the learned manifold to substitute for missing textual descriptions, effectively preserving the complex cross-modal logic and semantic integrity established during the supervised phase.
>
> 3.We sincerely appreciate your suggestion regarding the assessment of computational costs for practical deployment. To address this, we provide a detailed efficiency analysis in the table, comparing our hyperbolic manifold approach with standard Euclidean layers across four hierarchical levels. While hyperbolic transformations like exponential mappings and Möbius operations introduce higher mathematical complexity compared to linear Euclidean operations, this overhead remains entirely acceptable for practical applications. Within our architecture, these specific operations are executed only four times during a complete forward pass, occurring once per scale. Consequently, the absolute time added to the inference pipeline is minimal and negligible relative to the overall processing budget. This marginal increase in local computation represents a strategic trade-off that captures complex hierarchical relationships more effectively than Euclidean alternatives while maintaining a near-identical parameter footprint to ensure the model remains feasible for real-time deployment.
>
> Table 1. Assessment of computational costs for different deployment
> |Level (Scale)|Space Type|Params (M)|FLOPs (G)|Latency (ms)|
> |:---|:---|:---:|:---:|:---:|
> |Level 1|Euclidean|0.185536|0.087917|2.0082|
> ||Hyperbolic|0.185537|0.133926|35.8152|
> ||||||
> |Level 2|Euclidean|0.821248|0.087142|0.9992|
> ||Hyperbolic|0.821249|0.092896|4.0190|
> ||||||
> |Level 3|Euclidean|0.402688|0.087253|1.0011|
> ||Hyperbolic|0.402689|0.098758|3.0165|
> ||||||
> |Level 4|Euclidean|0.248704|0.087474|1.0009|
> ||Hyperbolic|0.248705|0.110480|4.0421|
>
> Key Questions:
> Please see the 'Weaknesses' sections above for evidence.

---

> > ### Author Rebuttal · Reviewer_Nzhe · 2026-04-02
> >
> > Thank you for the rebuttal and the additional experiments. The authors have addressed the primary concerns raised in the initial review.
> > As a minor formatting suggestion, referencing related points with explicit tags like "Q3 & W3" is usually clearer than a general statement like "Please see the 'Weaknesses' section above for evidence" This simple change helps reviewers easily track the exact correspondences.
> > I maintain my current rating for now and will finalize my recommendation after considering the overall feedback and discussions.

---

> > > ### Author Response · Authors · 2026-04-02
> > >
> > > Dear Reviewer,
> > >
> > > Thank you very much for acknowledging our rebuttal and the additional experiments! We are very glad that the main issues have been addressed. We completely agree with your suggestion regarding the formatting. Using explicit labels significantly improves clarity. As you suggested, we have reorganized our detailed responses below by combining the corresponding Weaknesses and Questions (e.g., "W1 & Q1") so that the one-to-one correspondence is clear and easy to track.
> > >
> > > Response to W1 & Q1:
> > >
> > > We sincerely thank you for this constructive feedback. Justifying hyperbolic geometry requires defining the hierarchical ontology inherent to infrared and visible image pairs. In our conceptual tree, textual semantics serve as the root node. Thermal prominence from the infrared modality acts as the parent node to define salient targets and the primary structural skeleton. Subordinate leaf nodes encompass high-frequency details and background textures from the visible modality. Since thermal targets logically dictate object presence and visible textures supply auxiliary attributes, this physical ontology necessitates hyperbolic space.
> > >
> > > Through our Text and Image Association Module, the textual embedding $\Psi_t$ acts as a semantic anchor near the origin of the Poincaré ball. The exponential expansion of hyperbolic space forces a natural semantic redistribution. Specifically, the network maps crucial thermal parent nodes closer to the textual origin due to their direct correlation with target descriptions, while pushing the vast array of visible texture child nodes into the expanding peripheral volume. This prevents squeezing fine-grained textures into a constrained representation space. Managed dynamically by the adaptive weighting term, the hyperbolic manifold fundamentally resolves the spatial disparity between singular thermal parent nodes and abundant visible texture child nodes.
> > >
> > > Response to W2 & Q2:
> > >
> > > Transferring semantic information from text to visual attributes relies on a dual mechanism of stochastic forced learning and manifold grounding. In ATGFM, a 30% text dropout probability acts as a supervision constraint, forcing shared projection layers to map explicit visual attributes into the latent space of their linguistic counterparts. Periodically withholding textual guidance compels the network to optimize attribute embeddings to become functionally congruent with text embeddings, ensuring the visual stream independently satisfies the fusion task's semantic requirements.
> > >
> > > The Text-Image Association Module (TIAM) further solidifies this semantic consistency. By projecting both modalities into a shared Poincaré ball, the hyperbolic loss ($\mathcal{L}_h$) enforces a relational hierarchy, grounding attributes to the semantic coordinates defined by the BLIP encoder. Since the hyperbolic manifold naturally preserves the hierarchical taxonomic structure of linguistic concepts, the attribute branch inherits these relational properties during training. At inference, attributes occupy these pre-established semantic coordinates even without explicit text. This allows the model to leverage the learned manifold to substitute for missing textual descriptions, effectively preserving the cross-modal logic and semantic integrity established during supervision.
> > >
> > > Response to W3 & Q3:
> > >
> > > We sincerely appreciate your suggestion to assess computational costs for practical deployment. To address this, Table 1 provides a detailed efficiency analysis comparing our hyperbolic manifold approach with standard Euclidean layers across four hierarchical levels. Although hyperbolic transformations like exponential mappings and Möbius operations introduce higher mathematical complexity, this overhead remains entirely acceptable. Within our architecture, these operations execute only once per scale, totaling four times during a complete forward pass. Consequently, the absolute time added to the inference pipeline is minimal and negligible relative to the overall budget. This marginal computational increase represents a strategic tradeoff. It captures complex hierarchical relationships more effectively than Euclidean alternatives while maintaining a near-identical parameter footprint to ensure real-time deployment feasibility.
> > >
> > > Table 1. Assessment of computational costs for different deployment
> > > |Level (Scale)|Space Type|Params (M)|FLOPs (G)|Latency (ms)|
> > > |---|---:|---:|---:|---:|
> > > |Level 1|Euclidean|0.185536|0.087917|2.0082|
> > > ||Hyperbolic|0.185537|0.133926|35.8152|
> > > ||||||
> > > |Level 2|Euclidean|0.821248|0.087142|0.9992|
> > > ||Hyperbolic|0.821249|0.092896|4.0190|
> > > ||||||
> > > |Level 3|Euclidean|0.402688|0.087253|1.0011|
> > > ||Hyperbolic|0.402689|0.098758|3.0165|
> > > ||||||
> > > |Level 4|Euclidean|0.248704|0.087474|1.0009|
> > > ||Hyperbolic|0.248705|0.110480|4.0421|
> > >
> > > We hope this revised format makes it much easier to track our responses. Thank you again for your constructive feedback, which has helped us improve both our manuscript and our response!

---

### Official Review · Reviewer_6WXe · 2026-03-11

**Soundness:** 3
**Presentation:** 3
**Significance:** 2
**Originality:** 3
**Overall Recommendation:** 4
**Confidence:** 4

**Summary:**

This paper proposes a multi-modal fusion framework TEDFusion, to fuse infrared and visible image. During training, the proposed Attribute-Text Gating Fusion Module extracts attribute features from infrared and visible images, and fuses them with text features through a gating mechanism. Meanwhile, the proposed Text-Image Association Module maps text and image features into hyperbolic space, where cross-modal semantic alignment is achieved through hyperbolic correlation learning and constraint loss. During test-time, the framework no longer relies on text input, but instead performs adaptive fusion of infrared and visible information solely based on the text prior learned during training.

**Compliance With Llm Reviewing Policy:**

Affirmed.

**Final Justification:**

The authors addressed most of my concerns.

**Key Questions For Authors:**

1. The authors need to directly verify the flattening phenomenon of language and visual features in Euclidean space claimed in this paper through specially designed experiments, rather than merely staying at the level of conceptual motivation.
2. The authors need to add clearer ablation experiments to distinguish whether the performance improvement mainly comes from the introduction of the text modality or from the hyperbolic space mapping mechanism proposed in this paper.
3. Since hyperbolic space is only used in the text-image related branch, while the subsequent fusion and reconstruction stages still mainly rely on Euclidean-space losses, the authors need to further clarify the practical role and positioning of hyperbolic modeling in the overall framework.
4. The authors need to supplement the sensitivity analysis of some manually designed parameters in ATGFM, such as the selection of low-level attributes, the setting of GLCM texture weights, and other key hyperparameters.
5. The authors need more convincing visualization or quantitative experiments to verify whether the hyperbolic space representation truly learns the hierarchical semantic structure from coarse to fine.

**Limitations:**

The authors should analyze the limitations of removing textual input during the testing stage.

**Strengths And Weaknesses:**

**Strengths**:
The authors align textual and visual features in hyperbolic space to alleviate the collapse caused by cross-modal feature fusion in Euclidean space, which is an interesting design. In addition, the proposed framework allows semantic information from the language modality to be fully utilized during training, while removing this branch during testing to avoid introducing additional deployment cost during inference. Moreover, the paper is relatively well written and experimentally comprehensive. The effectiveness of the infrared-visible fusion results is further validated through sufficient downstream experiments.

**Weaknesses**:
1. According to the ablation results in Figure 2, after the introduction of textual information, the performance gap between Euclidean space alignment and hyperbolic space alignment is not significant. Therefore, it is difficult to judge whether the final performance improvement is mainly due to the introduction of the text modality or the hyperbolic space mapping mechanism used in this paper. The authors need to further supplement more targeted experiments to verify the advantages of hyperbolic space modeling, and clarify its contribution relative to the performance gain brought by textual supervision.
2. The motivation of this paper is not sufficiently supported by convincing experiments showing that language and visual features will be flattened in Euclidean space. This perspective is interesting, but the authors should provide more analysis and design dedicated experiments to verify the existence of such flattening.
3. The authors discuss various limitations of Euclidean space. However, in the actual network design, hyperbolic space is only used in the module related to textual and visual features. In the subsequent image fusion and reconstruction stages, Euclidean-space losses based on pixel intensity, gradient, and SSIM are still adopted. Overall, the hyperbolic modeling does not achieve a unified geometric formulation for the entire fusion task.
4. In the ATGFM the authors manually select low-level features such as brightness, saturation, and contrast, and empirically assign the weights for GLCM texture. The authors should provide the corresponding parameter sensitivity analysis and further explain the choice of these low-level features.
5. The authors should verify through experiments whether the hyperbolic space truly learns the hierarchical semantics from coarse to fine through both visualization and quantitative analysis.

---

> ### Author Rebuttal · Authors · 2026-03-30
>
> Weakness:
> 1. We appreciate your comment. To clarify, we conducted an ablation using an iso-parametric Euclidean baseline, replacing hyperbolic components with standard Euclidean operators and Tanh functions to ensure equivalent nonlinear representation capacity.
> Table 1: Ablation study
> |Method|EN|SD|SF|AG|SCD|
> |:---|:---:|:---:|:---:|:---:|:---:|
> |Euclidean Baseline|7.01|43.57|7.62|3.14|1.82|
> |Ours|7.09|44.52|10.90|3.88|1.83|
> The Euclidean variant significantly underperforms our hyperbolic method across all key metrics, particularly in Spatial Frequency and Average Gradient, proving that adding text is insufficient for superior fusion if the latent space lacks the geometric properties required for hierarchical features. This demonstrates that our performance gains stem from the structural advantages of hyperbolic geometry in aligning diverse modalities rather than the text modality alone.
>
> 2. We appreciate your insight and validated the "feature flattening" phenomenon through spectral analysis of feature covariance matrices in both Euclidean and Hyperbolic spaces. Flattening occurs when features collapse into a low-dimensional space where variance concentrates in few principal components while others become redundant.
> Table 2. Feature Eigenvalue Distribution (Top 10)
> |Rank|Euclidean Eigenvalues|Hyperbolic Eigenvalues|
> |:---|:---:|:---:|
> |1|0.5417|0.2083|
> |2|0.4532|0.1752|
> |3|0.4283|0.1455|
> |4|0.1765|0.0669|
> |5|0.1408|0.0604|
> |6|0.1323|0.0536|
> |7|0.1177|0.0468|
> |8|0.1027|0.0380|
> |9|0.0754|0.0251|
> |10|0.0578|0.0199|
> As shown in Table 2, Euclidean eigenvalues follow a heavy-headed distribution where the initial three values are tenfold larger than the tenth, causing rapid spectral decay and a flattened representation that erases hierarchical detail, whereas Hyperbolic eigenvalues maintain a smoother and more balanced decay by distributing information equitably across dimensions to preserve fine-grained details, quantitatively proving that our Hyperbolic mapping prevents Euclidean feature flattening to enable richer multi-modal representations. The eigenvalue distribution figure is in (https://anonymous.4open.science/r/icml-2026-F735/review2-question2-eigenvalue%20distribution.png).
>
> These results provide direct quantitative evidence that our Hyperbolic mapping prevents the feature flattening observed in standard Euclidean frameworks, thereby allowing for a richer, more discriminative representation of multi-modal data.
>
> 3. Our framework integrates hyperbolic and Euclidean geometries to optimize semantic alignment and structural reconstruction. While hyperbolic geometry in the Poincaré ball effectively models hierarchical image-text relationships by using textual embeddings as structural anchors, Euclidean space remains essential for pixel-level texture preservation. This dual-geometry approach avoids the inherent conflicts of flat spaces when representing complex hierarchies. The hyperbolic branch informs the Euclidean pipeline by projecting aligned features via logarithmic mapping across multiple scales, providing a structured semantic prior that guides the network toward salient regions. This ensures the fused output is physically accurate and semantically consistent.
>
> 4. The four selected attributes form a comprehensive and orthogonal descriptor of multi-sensor scenes where S_u and B_r represent color and luminance while T_c and C_s capture structural order and intensity dispersion to ensure physical integrity. Sensitivity analysis (https://anonymous.4open.science/r/icml-2026-F735/review2-question4-sensitivity%20analysis%20of%20ATGFM.png) confirms the stability of Texture Complexity weighting: sweeping W_E from 0.1 to 0.9 yields a stable, linear trajectory without spikes or overfitting. This mathematical stability allows weight selection to be guided by physical sensor priors. Consequently, infrared weights emphasize the Energy term (0.7) to highlight thermal targets, while visible weights favor Homogeneity (0.4) for structural smoothness, resulting in a physically inspired and mathematically robust framework that generalizes stably across diverse scenarios.
>
> 5. We sincerely thank you for highlighting the importance of validating hierarchical semantic structures as this represents a vital challenge for hyperbolic embedding interpretability. While our results demonstrate clear performance gains, establishing a rigorous framework for visual verification of high-dimensional hierarchies remains an open research frontier requiring a dedicated, broader study to ensure scientific clarity. Consequently, we have pointed out this limitation in our revised manuscript and designated the geometric complexities of cross-modal hierarchy as a primary focus for our future research.
>
> Key Questions:
> Please see the 'Weaknesses' sections above for evidence.
>
> Limitation:
> Text-free inference simplifies usage, yet may slightly limit semantic precision in highly ambiguous scenes without explicit textual guidance.

---

> > ### Author Rebuttal · Reviewer_6WXe · 2026-04-05
> >
> > Through extensive experimental analysis, the authors addressed most of my concerns, and I adjusted my score to 4.

---

> > > ### Author Response · Authors · 2026-04-06
> > >
> > > We sincerely thank you for your time and for acknowledging our efforts in the revised manuscript. We are pleased that the new results met your expectations and appreciate the score adjustment.

---

### Official Review · Reviewer_p81v · 2026-03-13

**Soundness:** 3
**Presentation:** 2
**Significance:** 2
**Originality:** 2
**Overall Recommendation:** 4
**Confidence:** 4

**Summary:**

This work addresses the task of IVIF by proposing TEDFusion, a text-driven framework empowered by hyperbolic manifold learning. The authors argue that standard Euclidean spaces distort the hierarchical structure of visual and semantic data. Motivated by this, they utilize BLIP to extract textual anchors and propose projecting visual features and text embeddings into a shared hyperbolic space (a Poincare ball). The architecture relies on an Attribute Text Gating Fusion Module (ATGFM) to dynamically weight modalities and a Text-Image Association Module (TIAM) to perform Mobius translations for cross-modal alignment.

**Compliance With Llm Reviewing Policy:**

Affirmed.

**Final Justification:**

The authors have systematically addressed my core technical concerns, particularly by correcting the formulation of Eq. 2. I am happy to maintain my score to 4.

**Key Questions For Authors:**

- (*Method*) Does $[S_u, B_r, T_c, C_s]$   matches the $c,s,b,t$ in Figure 1?
- (*Method*)It's recommended that authors explain why the text is available in training while unavailable in inference
- (*Method*) I can see that the 2 modality images' features are processed and concatenated into a single one, waiting to be sent to TIAM. But what about the 2 text features? Are they also concatenated to get a single $X_t$?
- (*Experiment*) I don't understand why they trained only on MSRS and evaluated on TNO, RoadScene, and LLVIP without fine-tuning; that is a "Zero-Shot" generalisation test. But they don't explicitly claim this.
- (*Experiment*) Since the author states they use BLIP to generate the caption on the fly, what about other baselines? Do they use a stronger or weaker caption generator? It's hard to compare whether the caption or the caption generator is different. Moreover, since the method part states they operate in a text-free manner during inference, it is contradictory with "we use BLIP to automatically generate one caption per image on-the-fly during training and evaluation". Furthermore, I recommend that authors categorise the 11 baseline methods

**Limitations:**

yes

**Strengths And Weaknesses:**

**Strengths**
- The authors explore hyperbolic geometry (the Poincare ball) for IVIF. Attempting to address the volume limitations of Euclidean spaces for hierarchical semantic-visual associations is a mathematically creative approach that pushes the boundaries of traditional spatial-only fusion architectures.
- The experiments cover a robust set of scenarios. The authors evaluate their framework across four public datasets (TNO, RoadScene, MSRS, and LLVIP) and employ a rigorous suite of both full-reference and no-reference quantitative metrics (EN, SD, SF, AG, SCD) to assess fusion quality.

**Weakness**
- (*Introduction*) The authors state that Euclidean treatment is problematic for multimodal data, but the immediate evidence is the failure of image-text correlations.
- (*Figure 1*) What are $c,s,b,t$ inside the circle in Figure 1?
- (*Figure 1*) For BLIP extracted text, do they simply add them to the visual attribute in hyperbolic space with Mobius translation
- (*Method*) Figure 2 puts $I_{ir}$ and visible text at the top and connected, whereas $I_{vi}$ and the infrared image at the bottom are connected, which is wrong.
- (*Method*) There is an inconsistency: Figure 2(a) shows 2 separate ATGFM, whereas Figure 2(b) shows $X_{a_{ir}}, X_{a_{vi}}$ entering a single ATGFM, getting concatenated and passing through an MLP
- (*Method*) Figure 2(c) shows $X_v$ going directly into the $\mathrm{Exp}(\cdot)$ operation, which is contratictory to $\mathcal{P}_v = \mathrm{MLP}(\mathrm{Flatten}(X_v)) \in \mathbb{R}^{B \times d_h}$. More importantly, by applying the flattening operation before MLP and putting a flattened vector into Poincare ball, I think the authors contradict their motivation of "Euclidean space distorts low-level details", as image flattening averages all textures and edges.
- (*Method*) Eq 2 has a erorr that the numerator lacks $+ c\|-\Psi_t\|^2$ term. Without it, the translation operation is incorrect and results in a distorted gradient during back-prop.
- (*Experiment*) Author claims using Adam, but using standard Adam instead of Riemannian Adam in hyperbolic space may cause the math to collapse.

---

> ### Author Rebuttal · Authors · 2026-03-30
>
> Thank you very much for your valuable feedback. Our point-by-point responses are provided below:
>
> Weakness:
> 1. Flat Euclidean geometry imposes rigid distance metrics that distort cross-modal hierarchical semantics. In IVIF, text represents high-level global abstractions and visual attributes capture low-level local details. Euclidean space, with its polynomial volume growth, forces these multi-scale features to assume a uniform density, causing distinct semantic levels to become indistinguishable. Our method projects text semantics as root anchors and visual attributes as branches in the Poincaré ball. This hyperbolic mapping provides the geometric room to preserve parent-to-child hierarchies without distortion, bridging the cross-modal gap.
> 2. In Figure 1, $c, s, b, t$ represent contrast, saturation, brightness, and texture complexity features extracted from images (see Section 3.2).
> 3. Instead of adding text features directly to visual attributes, we project both embeddings into the Poincaré ball via an exponential map, where a Möbius translation aligns visual features relative to textual anchors. An adaptive hyperbolic loss evaluates cross-modal norm divergence to generate spatial weights, prioritizing critical regions. Minimizing the weighted Poincaré distance organizes visual attributes hierarchically around textual centers before returning to Euclidean space via a logarithmic map.
> 4. We apologize for the drafting error in Figure 2 and appreciate your observation. The modalities were indeed misaligned. We will correct these connections in the revised version to ensure I_{ir} and I_{vi} accurately correspond to their respective infrared and visible text descriptions.
> 5. We clarify that there is no architectural inconsistency. The multiple ATGFM blocks in Figure 2(a) are the exact same module applied independently across four multi-scale feature levels, and Figure 2(b) details its internal mechanism. The concatenation of X_{a_{ir}} and X_{a_{vi}} through a shared MLP in Figure 2(b) corresponds to the two converging gray arrows from both branches at each scale in Figure 2(a). This ensures dual-modality attributes are consistently fused at every hierarchical level.
> 6. We apologize for the imprecise use of "Flatten." We do not perform Global Average Pooling or collapse the image. Instead, "Flatten" denotes spatial unrolling into a sequence of tokens (reshaping from $B \times C \times H \times W$ to $B \times C \times N$, where $N = H \times W$), preserving local textures and edges. Each fine-grained feature independently undergoes linear projection and $\exp(\cdot)$ mapping into the Poincaré ball, aligning every local detail with the global text anchor without information loss. We will revise the notation to match Figure 2(c).
> 7. Thank you for pointing this out. You are correct that the $+ c\|-\Psi_t\|^2$ term is missing in the numerator of Equation 2. We apologize for this typographical oversight and will correct the formula in the revised version.
> 8. While Riemannian optimization updates manifold parameters, our learnable weights (MLPs, Transformers) remain in Euclidean space. Hyperbolic geometry applies strictly to features during the forward pass. Backpropagation gradients flow from the manifold back to the tangent space via the chain rule. Since the optimizer updates only these Euclidean weights, standard AdamW is mathematically sound. We use safe projections and gradient clamping to ensure stability and prevent collapse.
>
> Key Questions:
> 1. Yes, attributes [S_u, B_r, T_c, C_s] correspond to $s, b, t, c$ in Figure 1.
> 2. Text guides alignment and learns robust scene-attribute priors during training. During inference, the model operates text-free because the gating mechanism and learned priors enable visual attributes to substitute for textual guidance (see response to Reviewer #3, weakness 2).
> 3. Modality-specific text features h_{text1} and h_{text2} are concatenated into X_t to capture a full spectrum of descriptions, providing a robust anchor for the gating mechanism to guide vision-attribute alignment.
> 4. Superior performance on TNO, RoadScene, and LLVIP without fine-tuning highlights the framework's zero-shot generalization. The Poincaré ball’s negative curvature captures universal, modality-agnostic structural relationships and text-attribute correlations across datasets.
> 5. Unlike Text-IF and TextFusion which require manual annotations, our framework uses BLIP for automated training captions, maintaining text-free inference via learned priors. The 11 baselines categorize into: CNN-based (DenseFuse, U2Fusion), registration-focused (ReCoNet, SemLA), decomposition/generative (DeFusion, GIFuse, VDMUFusion, DCINN, GIFNet), and text-guided (Text-IF, TextFusion). TEDFusion achieves superior alignment and texture preservation without manual dependencies or rigid constraints.

---

> > ### Author Rebuttal · Reviewer_p81v · 2026-04-02
> >
> > I thank the authors for their detailed rebuttal, and I provide my point-to-point feedback below.
> >
> > > **W1, W3, & W8**
> >
> > I think the mathematical justification for using standard AdamW is technically sound now.
> >
> > > **W6**
> >
> > I think clarifying this as spatial unrolling effectively resolves my concern about the destruction of low-level textures. Unrolling into a sequence of tokens $B \times C \times N$ rather than applying GAP successfully preserves the local edges, ensuring the hyperbolic mapping actually has fine-grained details to organize hierarchically.
> >
> > > **W7**
> >
> > I am satisfied that adding the missing $+ c\|-\Psi_t\|^2$ to the numerator in Eq. 2. This correction is critical, as omitting it in the final manuscript would fundamentally distort the gradient calculation and reproducibility of the translation operation.
> >
> > > **W2, W4 & Q1**
> >
> > I thank the authors for acknowledging the modality misalignment in Figure 2 and confirming the exact variable mappings: $c,s,b,t$ to $S_u, B_r, T_c, C_s$.
> >
> > > **Q2, Q4, & Q5**
> >
> > I thank the authors for explicitly categorizing the 11 baselines and formally acknowledging the zero-shot nature of their evaluations on TNO, RoadScene, and LLVIP. I think the explanation of the learned priors adequately addresses the apparent contradiction regarding the use of BLIP captions.
> >
> > > **Final decision**
> >
> > The authors have systematically addressed my core technical concerns, particularly by correcting the formulation of Eq. 2. I am happy to maintain my score to 4.

---

> > > ### Author Response · Authors · 2026-04-02
> > >
> > > Thank you for acknowledging our rebuttal and raising the score. We will make sure to incorporate your valuable suggestions into the final version.

---

### Decision · Program_Chairs · 2026-04-30

**Decision:**

Accept (regular)

**Comment:**

The paper proposes TEDFusion, a text-driven framework for the fusion of the information from visible and IR images. To escape from the limitations of Euclidean geometry, it instead uses hyperbolic manifold learning to align textual and visual attributes during training. Text-free inference is applied at test time. The reviewers liked the ideas behind TEDFusion and appreciated their novelty. The authors also provided rich evaluation material, based on various dataset, multiple fusion metrics, and various tasks. The reviewers also appreciated the ablation studies, and the extra technical explanations provided in the rebuttal. Especially important given the topic of the paper, are the reviewers' concerns about the origin of the gains, i.e. whether it is truly the hyperbolic geometry that is responsible, rather than e.g. the textual supervision. The question was also raised whether the scene adaptation claim is justified. Also other issues were mentioned in the reviews, incl. reproducibility concerns and the computational overhead for hyperbolic computations. The authors successfully mitigated several in the rebuttals, through additional ablations, statistical tests, efficiency measurements, etc. All reviewers give a 'weak accept' score. I follow that consensus.